# On the Convergence of Stochastic Gradient MCMC Algorithms with High-Order Integrators

**Changyou Chen**[†]      **Nan Ding**[‡]      **Lawrence Carin**[†]

[†]Dept. of Electrical and Computer Engineering, Duke University, Durham, NC, USA

[‡]Google Inc., Venice, CA, USA

cchangyou@gmail.com; dingnan@google.com; lcarin@duke.edu

## Abstract

Recent advances in Bayesian learning with large-scale data have witnessed emergence of stochastic gradient MCMC algorithms (SG-MCMC), such as stochastic gradient Langevin dynamics (SGLD), stochastic gradient Hamiltonian MCMC (SGHMC), and the stochastic gradient thermostat. While finite-time convergence properties of the SGLD with a 1st-order Euler integrator have recently been studied, corresponding theory for general SG-MCMCs has not been explored. In this paper we consider general SG-MCMCs with high-order integrators, and develop theory to analyze finite-time convergence properties and their asymptotic invariant measures. Our theoretical results show faster convergence rates and more accurate invariant measures for SG-MCMCs with higher-order integrators. For example, with the proposed efficient 2nd-order symmetric splitting integrator, the *mean square error* (MSE) of the posterior average for the SGHMC achieves an optimal convergence rate of $L^{-4/5}$ at $L$ iterations, compared to $L^{-2/3}$ for the SGHMC and SGLD with 1st-order Euler integrators. Furthermore, convergence results of decreasing-step-size SG-MCMCs are also developed, with the same convergence rates as their fixed-step-size counterparts for a specific decreasing sequence. Experiments on both synthetic and real datasets verify our theory, and show advantages of the proposed method in two large-scale real applications.

## 1 Introduction

In large-scale Bayesian learning, diffusion based sampling methods have become increasingly popular. Most of these methods are based on Itô diffusions, defined as:

$$\mathrm{d}\,\mathbf{X}_t = F(\mathbf{X}_t)\mathrm{d}t + \sigma(\mathbf{X}_t)\mathrm{d}W_t \ . \tag{1}$$

Here $\mathbf{X}_t \in \mathbf{R}^n$ represents model states, $t$ the time index, $W_t$ is Brownian motion, functions $F : \mathbf{R}^n \to \mathbf{R}^n$ and $\sigma : \mathbf{R}^n \to \mathbf{R}^{n \times m}$ ($m$ not necessarily equal to $n$) are assumed to satisfy the usual Lipschitz continuity condition.

In a Bayesian setting, the goal is to design appropriate functions $F$ and $\sigma$, so that the stationary distribution, $\rho(\mathbf{X})$, of the Itô diffusion has a marginal distribution that is equal to the posterior distribution of interest. For example, 1st-order Langevin dynamics (LD) correspond to $\mathbf{X} = \boldsymbol{\theta}$, $F = -\nabla_{\boldsymbol{\theta}} U$ and $\sigma = \sqrt{2}\,\mathbf{I}_n$, with $\mathbf{I}_n$ being the $n \times n$ identity matrix; 2nd-order Langevin dynamics correspond to $\mathbf{X} = (\boldsymbol{\theta}, \mathbf{p})$, $F = \begin{pmatrix} \mathbf{p} \\ -D\,\mathbf{p} -\nabla_{\boldsymbol{\theta}} U \end{pmatrix}$, and $\sigma = \sqrt{2D}\begin{pmatrix} \mathbf{0} & \mathbf{0} \\ \mathbf{0} & \mathbf{I}_n \end{pmatrix}$ for some $D > 0$. Here $U$ is the unnormalized negative log-posterior, and $\mathbf{p}$ is known as the momentum [1, 2]. Based on the Fokker-Planck equation [3], the stationary distributions of these dynamics exist and their marginals over $\boldsymbol{\theta}$ are equal to $\rho(\boldsymbol{\theta}) \propto \exp(-U(\boldsymbol{\theta}))$, the posterior distribution we are interested in.

Since Itô diffusions are continuous-time Markov processes, exact sampling is in general infeasible. As a result, the following two approximations have been introduced in the machine learning liter-

ature [1, 2, 4], to make the sampling numerically feasible and practically scalable: 1) Instead of analytically integrating infinitesimal increments $\mathrm{d}t$, numerical integration over small step $h$ is used to approximate the integration of the true dynamics. Although many numerical schemes have been studied in the SDE literature, in machine learning only the 1st-order Euler scheme is widely applied. 2) During every integration, instead of working with the gradient of the full negative log-posterior $U(\boldsymbol{\theta})$, a stochastic-gradient version of it, $\tilde{U}_l(\boldsymbol{\theta})$, is calculated from the $l$-th minibatch of data, important when considering problems with massive data. In this paper, we call algorithms based on 1) and 2) SG-MCMC algorithms. To be complete, some recently proposed SG-MCMC algorithms are briefly reviewed in Appendix A. SG-MCMC algorithms often work well in practice, however some theoretical concerns about the convergence properties have been raised [5–7].

Recently, [5, 6, 8] showed that the SGLD [4] converges weakly to the true posterior. In [7], the author studied the sample-path inconsistency of the Hamiltonian PDE with stochastic gradients (but not the SGHMC), and pointed out its incompatibility with data subsampling. However, real applications only require convergence in the weak sense, *i.e.*, instead of requiring sample-wise convergence as in [7], only laws of sample paths are of concern*. Very recently, the invariance measure of an SG-MCMC with a specific stochastic gradient noise was studied in [9]. However, the technique is not readily applicable to our general setting.

In this paper we focus on general SG-MCMCs, and study the role of their numerical integrators. Our main contributions include: *i*) From a theoretical viewpoint, we prove weak convergence results for general SG-MCMCs, which are of practical interest. Specifically, for a $K$th-order numerical integrator, the bias of the expected sample average of an SG-MCMC at iteration $L$ is upper bounded by $L^{-K/(K+1)}$ with optimal step size $h \propto L^{-1/(K+1)}$, and the MSE by $L^{-2K/(2K+1)}$ with optimal $h \propto L^{-1/(2K+1)}$. This generalizes the results of the SGLD with an Euler integrator ($K = 1$) in [5, 6, 8], and is better when $K \geq 2$; *ii*) From a practical perspective, we introduce a numerically efficient 2nd-order integrator, based on symmetric splitting schemes [9]. When applied to the SGHMC, it outperforms existing algorithms, including the SGLD and SGHMC with Euler integrators, considering both synthetic and large real datasets.

## 2 Preliminaries & Two Approximation Errors in SG-MCMCs

In weak convergence analysis, instead of working directly with sample-paths in (1), we study how the expected value of any suitably smooth statistic of $\mathbf{X}_t$ evolves in time. This motivates the introduction of an (infinitesimal) *generator*. Formally, the *generator* $\mathcal{L}$ of the diffusion (1) is defined for any compactly supported twice differentiable function $f : \mathbf{R}^n \to \mathbf{R}$, such that,

$$\mathcal{L}f(\mathbf{X}_t) \triangleq \lim_{h \to 0^+} \frac{\mathbb{E}\left[f(\mathbf{X}_{t+h})\right] - f(\mathbf{X}_t)}{h} = \left( F(\mathbf{X}_t) \cdot \nabla + \frac{1}{2} \left( \sigma(\mathbf{X}_t)\sigma(\mathbf{X}_t)^T \right) : \nabla \nabla^T \right) f(\mathbf{X}_t) ,$$

where $\mathbf{a} \cdot \mathbf{b} \triangleq \mathbf{a}^T \mathbf{b}$, $\mathbf{A} : \mathbf{B} \triangleq \operatorname{tr}(\mathbf{A}^T \mathbf{B})$, $h \to 0^+$ means $h$ approaches zero along the positive real axis. $\mathcal{L}$ is associated with an integrated form via Kolmogorov's backward equation†: $\mathbb{E}\left[f(\mathbf{X}_T^e)\right] = e^{T\mathcal{L}}f(\mathbf{X}_0)$, where $\mathbf{X}_T^e$ denotes the exact solution of the diffusion (1) at time $T$. The operator $e^{T\mathcal{L}}$ is called the Kolmogorov operator for the diffusion (1). Since diffusion (1) is continuous, it is generally infeasible to solve analytically (so is $e^{T\mathcal{L}}$). In practice, a local numerical integrator is used for every small step $h$, with the corresponding Kolmogorov operator $P_h$ approximating $e^{h\mathcal{L}}$. Let $\mathbf{X}_{lh}^n$ denote the approximate sample path from such a **n**umerical integrator; similarly, we have $\mathbb{E}[f(\mathbf{X}_{lh}^n)] = P_h f(\mathbf{X}_{(l-1)h}^n)$. Let $\mathcal{A} \circ \mathcal{B}$ denote the *composition* of two operators $\mathcal{A}$ and $\mathcal{B}$, *i.e.*, $\mathcal{A}$ is evaluated on the output of $\mathcal{B}$. For time $T = Lh$, we have the following approximation

$$\mathbb{E}\left[f(\mathbf{X}_T^e)\right] \stackrel{A_1}{=} e^{h\mathcal{L}} \circ \ldots \circ e^{h\mathcal{L}}f(\mathbf{X}_0) \stackrel{A_2}{\approx} P_h \circ \ldots \circ P_h f(\mathbf{X}_0) = \mathbb{E}[f(\mathbf{X}_T^n)],$$

with $L$ compositions, where $A_1$ is obtained by decomposing $T\mathcal{L}$ into $L$ sub-operators, each for a minibatch of data, while approximation $A_2$ is manifested by approximating the infeasible $e^{h\mathcal{L}}$ with $P_h$ from a feasible integrator, *e.g.*, the symmetric splitting integrator proposed later, such that

$\mathbb{E}\left[f(\mathbf{X}_T^n)\right]$ is close to the exact expectation $\mathbb{E}\left[f(\mathbf{X}_T^e)\right]$. The latter is the first approximation error introduced in SG-MCMCs. Formally, to characterize the degree of approximation accuracies for different numerical methods, we use the following definition.

**Definition 1.** *An integrator is said to be a $K$th-order local integrator if for any smooth and bounded function $f$, the corresponding Kolmogorov operator $P_h$ satisfies the following relation:*

$$P_h f(\mathbf{x}) = e^{h\mathcal{L}} f(\mathbf{x}) + O(h^{K+1}) . \tag{2}$$

The second approximation error is manifested when handling large data. Specifically, the SGLD and SGHMC use stochastic gradients in the 1st and 2nd-order LDs, respectively, by replacing in $F$ and $\mathcal{L}$ the full negative log-posterior $U$ with a scaled log-posterior, $\tilde{U}_l$, from the $l$-th minibatch. We denote the corresponding generators with stochastic gradients as $\tilde{\mathcal{L}}_l$, *e.g.*, the generator in the $l$-th minibatch for the SGHMC becomes $\tilde{\mathcal{L}}_l = \mathcal{L} + \Delta V_l$, where $\Delta V_l = (\nabla_\theta \tilde{U}_l - \nabla_\theta U) \cdot \nabla_p$. As a result, in SG-MCMC algorithms, we use the noisy operator $\tilde{P}_h^l$ to approximate $e^{h\tilde{\mathcal{L}}_l}$ such that $\mathbb{E}[f(\mathbf{X}_{lh}^{n,s})] = \tilde{P}_h^l f(\mathbf{X}_{(l-1)h})$, where $\mathbf{X}_{lh}^{n,s}$ denotes the **n**umerical sample-path with **s**tochastic gradient noise, *i.e.*,

$$\mathbb{E}\left[f(\mathbf{X}_T^e)\right] \overset{B_1}{\simeq} e^{h\tilde{\mathcal{L}}_L} \circ \ldots \circ e^{h\tilde{\mathcal{L}}_1} f(\mathbf{X}_0) \overset{B_2}{\simeq} \tilde{P}_h^L \circ \ldots \circ \tilde{P}_h^1 f(\mathbf{X}_0) = \mathbb{E}[f(\mathbf{X}_T^{n,s})]. \tag{3}$$

Approximations $B_1$ and $B_2$ in (3) are from the *stochastic gradient* and *numerical integrator* approximations, respectively. Similarly, we say $\tilde{P}_h^l$ corresponds to a $K$th-order local integrator of $\tilde{\mathcal{L}}_l$ if $\tilde{P}_h^l f(\mathbf{x}) = e^{h\tilde{\mathcal{L}}_l} f(\mathbf{x}) + O(h^{K+1})$. In the following sections, we focus on SG-MCMCs which use numerical integrators with stochastic gradients, and for the first time analyze how the two introduced errors affect their convergence behaviors. For notational simplicity, we henceforth use $\mathbf{X}_t$ to represent the approximate sample-path $\mathbf{X}_t^{n,s}$.

## 3 Convergence Analysis

This section develops theory to analyze finite-time convergence properties of general SG-MCMCs with both fixed and decreasing step sizes, as well as their asymptotic invariant measures.

### 3.1 Finite-time error analysis

Given an ergodic[‡] Itô diffusion (1) with an invariant measure $\rho(\mathbf{x})$, the posterior average is defined as: $\bar{\phi} \triangleq \int_{\mathcal{X}} \phi(\mathbf{x}) \rho(\mathbf{x}) \mathrm{d}\mathbf{x}$ for some test function $\phi(\mathbf{x})$ of interest. For a given numerical method with generated samples $(\mathbf{X}_{lh})_{l=1}^L$, we use the *sample average* $\hat{\phi}$ defined as $\hat{\phi} = \frac{1}{L} \sum_{l=1}^L \phi(\mathbf{X}_{lh})$ to approximate $\bar{\phi}$. In the analysis, we define a functional $\psi$ that solves the following *Poisson Equation*:

$$\mathcal{L}\psi(\mathbf{X}_{lh}) = \phi(\mathbf{X}_{lh}) - \bar{\phi}, \text{ or equivalently, } \frac{1}{L}\sum_{l=1}^L \mathcal{L}\psi(\mathbf{X}_{lh}) = \hat{\phi} - \bar{\phi}. \tag{4}$$

The solution functional $\psi(\mathbf{X}_{lh})$ characterizes the difference between $\phi(\mathbf{X}_{lh})$ and the posterior average $\bar{\phi}$ for every $\mathbf{X}_{lh}$, thus would typically possess a unique solution, which is at least as smooth as $\phi$ under the elliptic or hypoelliptic settings [12]. In the unbounded domain of $\mathbf{X}_{lh} \in \mathbf{R}^n$, to make the presentation simple, we follow [6] and make certain assumptions on the solution functional, $\psi$, of the Poisson equation (4), which are used in the detailed proofs. Extensive empirical results have indicated the assumptions to hold in many real applications, though extra work is needed for theoretical verifications for different models, which is beyond the scope of this paper.

**Assumption 1.** *$\psi$ and its up to 3rd-order derivatives, $\mathcal{D}^k\psi$, are bounded by a function[§] $\mathcal{V}$, i.e., $\|\mathcal{D}^k\psi\| \le C_k \mathcal{V}^{p_k}$ for $k = (0, 1, 2, 3)$, $C_k, p_k > 0$. Furthermore, the expectation of $\mathcal{V}$ on $\{\mathbf{X}_{lh}\}$ is bounded: $\sup_l \mathbb{E}\mathcal{V}^p(\mathbf{X}_{lh}) < \infty$, and $\mathcal{V}$ is smooth such that $\sup_{s \in (0,1)} \mathcal{V}^p(s\mathbf{X} + (1-s)\mathbf{Y}) \le C(\mathcal{V}^p(\mathbf{X}) + \mathcal{V}^p(\mathbf{Y}))$, $\forall \mathbf{X}, \mathbf{Y}, p \le \max\{2p_k\}$ for some $C > 0$.*

---

[‡]See [6, 11] for conditions to ensure (1) is ergodic.

[§]The existence of such function can be translated into finding a Lyapunov function for the corresponding SDEs, an important topic in PDE literatures [13]. See Assumption 4.1 in [6] and Appendix C for more details.

We emphasize that our proof techniques are related to those of the SGLD [6, 12], but with significant distinctions in that, instead of expanding the function $\psi(\mathbf{X}_{lh})$ [6], whose parameter $\mathbf{X}_{lh}$ does not endow an explicit form in general SG-MCMCs, we start from expanding the Kolmogorov's backward equation for each minibatch. Moreover, our techniques apply for general SG-MCMCs, instead of for one specific algorithm. More specifically, given a $K$th-order local integrator with the corresponding Kolmogorov operator $\tilde{P}_h^l$, according to Definition 1 and (3), the Kolmogorov's backward equation for the $l$-th minibatch can be expanded as:

$$\mathbb{E}[\psi(\mathbf{X}_{lh})] = \tilde{P}_h^l \psi(\mathbf{X}_{(l-1)h}) = e^{h\tilde{\mathcal{L}}_l}\psi(\mathbf{X}_{(l-1)h}) + O(h^{K+1})$$

$$= \left(\mathbb{I} + h\tilde{\mathcal{L}}_l\right)\psi(\mathbf{X}_{(l-1)h}) + \sum_{k=2}^{K}\frac{h^k}{k!}\tilde{\mathcal{L}}_l^k\psi(\mathbf{X}_{(l-1)h}) + O(h^{K+1}),\qquad(5)$$

where $\mathbb{I}$ is the identity map. Recall that $\tilde{\mathcal{L}}_l = \mathcal{L} + \Delta V_l$, $e.g.$, $\Delta V_l = (\nabla_\theta \tilde{U}_l - \nabla_\theta U)\cdot\nabla_p$ in SGHMC. By further using the Poisson equation (4) to simplify related terms associated with $\mathcal{L}$, after some algebra shown in Appendix D, the bias can be derived from (5) as: $|\mathbb{E}\hat{\phi} - \bar{\phi}| =$

$$\left|\frac{\mathbb{E}[\psi(\mathbf{X}_{lh})] - \psi(\mathbf{X}_0)}{Lh} - \frac{1}{L}\sum_l \mathbb{E}[\Delta V_l\psi(\mathbf{X}_{(l-1)h})] - \sum_{k=2}^{K}\frac{h^{k-1}}{k!L}\sum_{l=1}^{L}\mathbb{E}[\tilde{\mathcal{L}}_l^k\psi(\mathbf{X}_{(l-1)h})]\right| + O(h^K).$$

All terms in the above equation can be bounded, with details provided in Appendix D. This gives us a bound for the bias of an SG-MCMC algorithm in Theorem 2.

**Theorem 2.** *Under Assumption 1, let $\|\cdot\|$ be the operator norm. The bias of an SG-MCMC with a $K$th-order integrator at time $T = hL$ can be bounded as:*

$$\left|\mathbb{E}\hat{\phi} - \bar{\phi}\right| = O\left(\frac{1}{Lh} + \frac{\sum_l \|\mathbb{E}\Delta V_l\|}{L} + h^K\right).$$

Note the bound above includes the term $\sum_l \|\mathbb{E}\Delta V_l\|/L$, measuring the difference between the expectation of stochastic gradients and the true gradient. It vanishes when the stochastic gradient is an unbiased estimation of the exact gradient, an assumption made in the SGLD. This on the other hand indicates that if the stochastic gradient is biased, $|\mathbb{E}\hat{\phi} - \bar{\phi}|$ might diverge when the growth of $\sum_l \|\mathbb{E}\Delta V_l\|$ is faster than $O(L)$. We point this out to show our result to be more informative than that of the SGLD [6], though this case might not happen in real applications. By expanding the proof for the bias, we are also able to bound the MSE of SG-MCMC algorithms, given in Theorem 3.

**Theorem 3.** *Under Assumption 1, and assume $\tilde{U}_l$ is an unbiased estimate of $U_l$. For a smooth test function $\phi$, the MSE of an SG-MCMC with a $K$th-order integrator at time $T = hL$ is bounded, for some $C > 0$ independent of $(L, h)$, as*

$$\mathbb{E}\left(\hat{\phi} - \bar{\phi}\right)^2 \leq C\left(\frac{\frac{1}{L}\sum_l \mathbb{E}\|\Delta V_l\|^2}{L} + \frac{1}{Lh} + h^{2K}\right).$$

Compared to the SGLD [6], the extra term $\frac{1}{L^2}\sum_l \mathbb{E}\|\Delta V_l\|^2$ relates to the variance of noisy gradients. As long as the variance is bounded, the MSE still converges with the same rate. Specifically, when optimizing bounds for the bias and MSE, the optimal bias decreases at a rate of $L^{-K/(K+1)}$ with step size $h \propto L^{-1/(K+1)}$; while this is $L^{-2K/(2K+1)}$ with step size $h \propto L^{-1/(2K+1)}$ for the MSE[¶]. These rates decrease faster than those of the SGLD [6] when $K \geq 2$. The case of $K = 2$ for the SGHMC with our proposed symmetric splitting integrator is discussed in Section 4.

### 3.2 Stationary invariant measures

The asymptotic invariant measures of SG-MCMCs correspond to $L$ approaching infinity in the above analysis. According to the bias and MSE above, asymptotically ($L \to \infty$) the sample average $\hat{\phi}$ is a random variable with mean $\mathbb{E}\hat{\phi} = \bar{\phi} + O(h^K)$, and variance $\mathbb{E}(\hat{\phi} - \mathbb{E}\hat{\phi})^2 \leq \mathbb{E}(\hat{\phi} - \bar{\phi})^2 + \mathbb{E}(\bar{\phi} - \mathbb{E}\hat{\phi})^2 = O(h^{2K})$, close to the true $\bar{\phi}$. This section defines distance between measures, and studies more formally how the approximation errors affect the invariant measures of SG-MCMC algorithms.

---

[¶]To compare with the standard MCMC convergence rate of $1/2$, the rate needs to be taken a square root.

First we note that under mild conditions, the existence of a stationary invariant measure for an SG-MCMC can be guaranteed by application of the Krylov–Bogolyubov Theorem [14]. Examining the conditions is beyond the scope of this paper. For simplicity, we follow [12] and assume stationary invariant measures do exist for SG-MCMCs. We denote the corresponding invariant measure as $\tilde{\rho}_h$, and the true posterior of a model as $\rho$. Similar to [12], we assume our numerical solver is geometric ergodic, meaning that for a test function $\phi$, we have $\int_{\mathcal{X}} \phi(\mathbf{x})\tilde{\rho}_h(\mathrm{d}\,\mathbf{x}) = \int_{\mathcal{X}} \mathbb{E}_{\mathbf{x}}\phi(\mathbf{X}_{lh})\tilde{\rho}_h(\mathrm{d}\,\mathbf{x})$ for any $l \geq 0$ from the ergodic theorem, where $\mathbb{E}_{\mathbf{x}}$ denotes the expectation conditional on $\mathbf{X}_0 = \mathbf{x}$. The geometric ergodicity implies that the integration is independent of the starting point of an algorithm. Given this, we have the following theorem on invariant measures of SG-MCMCs.

**Theorem 4.** *Assume that a $K$th-order integrator is geometric ergodic and its invariance measures $\tilde{\rho}_h$ exist. Define the distance between the invariant measures $\tilde{\rho}_h$ and $\rho$ as: $d(\tilde{\rho}_h, \rho) \triangleq \sup_\phi \left| \int_{\mathcal{X}} \phi(\mathbf{x})\tilde{\rho}_h(\mathrm{d}\,\mathbf{x}) - \int_{\mathcal{X}} \phi(\mathbf{x})\rho(\mathrm{d}\,\mathbf{x}) \right|$. Then any invariant measure $\tilde{\rho}_h$ of an SG-MCMC is close to $\rho$ with an error up to an order of $O(h^K)$, i.e., there exists some $C \geq 0$ such that: $d(\tilde{\rho}_h, \rho) \leq Ch^K$.*

For a $K$th-order integrator with full gradients, the corresponding invariant measure has been shown to be bounded by an order of $O(h^K)$ [9, 12]. As a result, Theorem 4 suggests only orders of numerical approximations but not the stochastic gradient approximation affect the asymptotic invariant measure of an SG-MCMC algorithm. This is also reflected by experiments presented in Section 5.

### 3.3 SG-MCMCs with decreasing step sizes

The original SGLD was first proposed with a decreasing-step-size sequence [4], instead of fixing step sizes, as analyzed in [6]. In [5], the authors provide theoretical foundations on its asymptotic convergence properties. We demonstrate in this section that for general SG-MCMC algorithms, decreasing step sizes for each minibatch are also feasible. Note our techniques here are different from those used for the decreasing-step-size SGLD [5], which interestingly result in similar convergence patterns. Specifically, by adapting the same techniques used in the previous sections, we establish conditions on the step size sequence to ensure asymptotic convergence, and develop theory on their finite-time ergodic error as well. To guarantee asymptotic consistency, the following conditions on decreasing step size sequences are required.

**Assumption 2.** *The step sizes $\{h_l\}$ are decreasing[||], i.e., $0 < h_{l+1} < h_l$, and satisfy that 1) $\sum_{l=1}^{\infty} h_l = \infty$; and 2) $\lim_{L\to\infty} \frac{\sum_{l=1}^{L} h_l^{K+1}}{\sum_{l=1}^{L} h_l} = 0$.*

Denote the finite sum of step sizes as $S_L \triangleq \sum_{l=1}^{L} h_l$. Under Assumption 2, we need to modify the sample average $\bar{\phi}$ defined in Section 3.1 as a weighted summation of $\{\phi(\mathbf{X}_{lh})\}$: $\tilde{\phi} = \sum_{l=1}^{L} \frac{h_l}{S_L} \phi(\mathbf{X}_{lh})$. For simplicity, we assume $\tilde{U}_l$ to be an unbiased estimate of $U$ such that $\mathbb{E}\Delta V_l = 0$. Extending techniques in previous sections, we develop the following bounds for the bias and MSE.

**Theorem 5.** *Under Assumptions 1 and 2, for a smooth test function $\phi$, the bias and MSE of a decreasing-step-size SG-MCMC with a $K$th-order integrator at time $S_L$ are bounded as:*

$$\text{BIAS: } \left| \mathbb{E}\tilde{\phi} - \bar{\phi} \right| = O\left( \frac{1}{S_L} + \frac{\sum_{l=1}^{L} h_l^{K+1}}{S_L} \right) \tag{6}$$

$$\text{MSE: } \mathbb{E}\left( \tilde{\phi} - \bar{\phi} \right)^2 \leq C\left( \sum_l \frac{h_l^2}{S_L^2} \mathbb{E}\|\Delta V_l\|^2 + \frac{1}{S_L} + \frac{(\sum_{l=1}^{L} h_l^{K+1})^2}{S_L^2} \right). \tag{7}$$

*As a result, the asymptotic bias approaches 0 according to the assumptions. If further assuming[**] $\frac{\sum_{l=1}^{\infty} h_l^2}{S_L^2} = 0$, the MSE also goes to 0. In words, the decreasing-step-size SG-MCMCs are consistent.*

Among the kinds of decreasing step size sequences, a commonly recognized one is $h_l \propto l^{-\alpha}$ for $0 < \alpha < 1$. We show in the following corollary that such a sequence leads to a valid sequence.

**Corollary 6.** *Using the step size sequences $h_l \propto l^{-\alpha}$ for $0 < \alpha < 1$, all the step size assumptions in Theorem 5 are satisfied. As a result, the bias and MSE approach zero asymptotically, i.e., the sample average $\tilde{\phi}$ is asymptotically consistent with the posterior average $\bar{\phi}$.*

---

[||]Actually the sequence need not be decreasing; we assume it is decreasing for simplicity.

[**]The assumption of $\sum_{l=1}^{\infty} h_l^2 < \infty$ satisfies this requirement, but is weaker than the original assumption.

**Remark 7.** *Theorem 5 indicates the sample average $\tilde{\phi}$ asymptotically converges to the true posterior average $\bar{\phi}$. It is possible to find out the optimal decreasing rates for the specific decreasing sequence $h_l \propto l^{-\alpha}$. Specifically, using the bounds for $\sum_{l=1}^{L} l^{-\alpha}$ (see the proof of Corollary 6), for the two terms in the bias (6) in Theorem 5, $\frac{1}{S_L}$ decreases at a rate of $O(L^{\alpha-1})$, whereas $(\sum_{l=1}^{L} h_l^{K+1})/S_L$ decreases as $O(L^{-K\alpha})$. The balance between these two terms is achieved when $\alpha = 1/(K+1)$, which agrees with Theorem 2 on the optimal rate of fixed-step-size SG-MCMCs. Similarly, for the MSE (7), the first term decreases as $L^{-1}$, independent of $\alpha$, while the second and third terms decrease as $O(L^{\alpha-1})$ and $O(L^{-2K\alpha})$, respectively, thus the balance is achieved when $\alpha = 1/(2K+1)$, which also agrees with the optimal rate for the fixed-step-size MSE in Theorem 3.*

According to Theorem 5, one theoretical advantage of decreasing-step-size SG-MCMCs over fixed-step-size variants is the asymptotically unbiased estimation of posterior averages, though the benefit might not be significant in large-scale real applications where the asymptotic regime is not reached.

## 4 Practical Numerical Integrators

Given the theory for SG-MCMCs with high-order integrators, we here propose a 2nd-order symmetric splitting integrator for practical use. The Euler integrator is known as a 1st-order integrator; the proof and its detailed applications on the SGLD and SGHMC are given in Appendix I.

The main idea of the symmetric splitting scheme is to split the local generator $\tilde{\mathcal{L}}_l$ into several sub-generators that can be solved analytically[††]. Unfortunately, one cannot easily apply a splitting scheme with the SGLD. However, for the SGHMC, it can be readily split into: $\tilde{\mathcal{L}}_l = \mathcal{L}_A + \mathcal{L}_B + \mathcal{L}_{O_l}$, where

$$\mathcal{L}_A = \mathbf{p} \cdot \nabla_{\boldsymbol{\theta}}, \quad \mathcal{L}_B = -D\,\mathbf{p} \cdot \nabla_{\mathbf{p}}, \quad \mathcal{L}_{O_l} = -\nabla_{\boldsymbol{\theta}} \tilde{U}(\boldsymbol{\theta}) \cdot \nabla_{\mathbf{p}} + 2D\,\mathbf{I}_n : \nabla_{\mathbf{p}} \nabla_{\mathbf{p}}^T. \tag{8}$$

These sub-generators correspond to the following SDEs, which are all analytically solvable:

$$A: \begin{cases} \mathrm{d}\boldsymbol{\theta} &= \mathbf{p}\,\mathrm{d}t \\ \mathrm{d}\,\mathbf{p} &= 0 \end{cases}, B: \begin{cases} \mathrm{d}\boldsymbol{\theta} &= 0 \\ \mathrm{d}\,\mathbf{p} &= -D\,\mathbf{p}\,\mathrm{d}t \end{cases}, O: \begin{cases} \mathrm{d}\boldsymbol{\theta} &= 0 \\ \mathrm{d}\,\mathbf{p} &= -\nabla_{\boldsymbol{\theta}} \tilde{U}_l(\boldsymbol{\theta})\mathrm{d}t + \sqrt{2D}\mathrm{d}W \end{cases} \tag{9}$$

Based on these sub-SDEs, the local Kolmogorov operator $\tilde{P}_h^l$ is defined as:

$$\mathbb{E}[f(\mathbf{X}_{lh})] = \tilde{P}_h^l f(\mathbf{X}_{(l-1)h}), \quad \text{where,} \quad \tilde{P}_h^l \triangleq e^{\frac{h}{2}\mathcal{L}_A} \circ e^{\frac{h}{2}\mathcal{L}_B} \circ e^{h\mathcal{L}_{O_l}} \circ e^{\frac{h}{2}\mathcal{L}_B} \circ e^{\frac{h}{2}\mathcal{L}_A},$$

so that the corresponding updates for $\mathbf{X}_{lh} = (\boldsymbol{\theta}_{lh}, \mathbf{p}_{lh})$ consist of the following 5 steps:

$$\boldsymbol{\theta}_{lh}^{(1)} = \boldsymbol{\theta}_{(l-1)h} + \mathbf{p}_{(l-1)h}\,h/2 \Rightarrow \mathbf{p}_{lh}^{(1)} = e^{-Dh/2}\,\mathbf{p}_{(l-1)h} \Rightarrow \mathbf{p}_{lh}^{(2)} = \mathbf{p}_{lh}^{(1)} - \nabla_{\boldsymbol{\theta}} \tilde{U}_l(\boldsymbol{\theta}_{lh}^{(1)})h + \sqrt{2Dh}\boldsymbol{\zeta}_l$$

$$\Rightarrow \mathbf{p}_{lh} = e^{-Dh/2}\mathbf{p}_{lh}^{(2)} \Rightarrow \boldsymbol{\theta}_{lh} = \boldsymbol{\theta}_{lh}^{(1)} + \mathbf{p}_{lh}\,h/2,$$

where $(\boldsymbol{\theta}_{lh}^{(1)}, \mathbf{p}_{lh}^{(1)}, \mathbf{p}_{lh}^{(2)})$ are intermediate variables. We denote such a splitting method as the ABOBA scheme. From the Markovian property of a Kolmogorov operator, it is readily seen that all such symmetric splitting schemes (with different orders of 'A', 'B' and 'O') are equivalent [15]. Lemma 8 below shows the symmetric splitting scheme is a 2nd-order local integrator.

**Lemma 8.** *The symmetric splitting scheme is a 2nd-order local integrator, i.e., the corresponding Kolmogorov operator $\tilde{P}_h^l$ satisfies: $\tilde{P}_h^l = e^{h\tilde{\mathcal{L}}_l} + O(h^3)$.*

When this integrator is applied to the SGHMC, the following properties can be obtained.

**Remark 9.** *Applying Theorem 2 to the SGHMC with the symmetric splitting scheme ($K = 2$), the bias is bounded as: $|\mathbb{E}\hat{\phi} - \bar{\phi}| = O(\frac{1}{Lh} + \frac{\sum_l \|\mathbb{E}\Delta V_l\|}{L} + h^2)$. The optimal bias decreasing rate is $L^{-2/3}$, compared to $L^{-1/2}$ for the SGLD [6]. Similarly, the MSE is bounded by: $\mathbb{E}(\hat{\phi} - \bar{\phi})^2 \leq C(\frac{\frac{1}{L}\sum_l \mathbb{E}\|\Delta V_l\|^2}{L} + \frac{1}{Lh} + h^4)$, decreasing optimally as $L^{-4/5}$ with step size $h \propto L^{-1/5}$, compared to the MSE of $L^{-2/3}$ for the SGLD [6]. This indicates that the SGHMC with the splitting integrator converges faster than the SGLD and SGHMC with 1st-order Euler integrators.*

**Remark 10.** *For a decreasing-step-size SGHMC, based on Remark 7, the optimal step size decreasing rate for the bias is $\alpha = 1/3$, and $\alpha = 1/5$ for the MSE. These agree with their fixed-step-size counterparts in Remark 9, thus are faster than the SGLD/SGHMC with 1st-order Euler integrators.*

---

[††]This is different from the traditional splitting in SDE literatures[9, 15], where $\mathcal{L}$ instead of $\tilde{\mathcal{L}}_l$ is split.

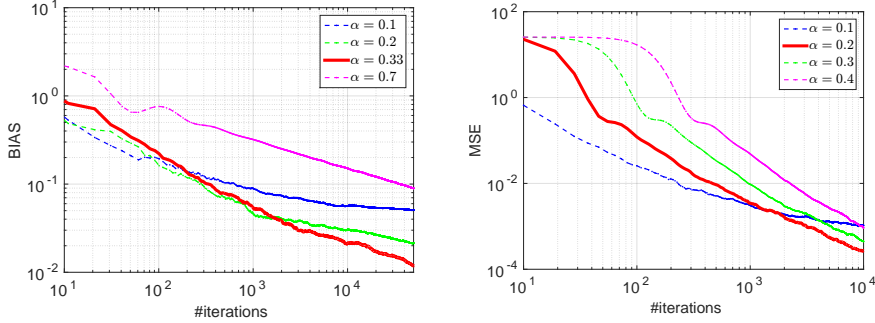

Figure 2: *Bias* of SGHMC-D (left) and *MSE* of SGHMC-F (right) with different step size rates $\alpha$. Thick red curves correspond to theoretically optimal rates.

## 5 Experiments

We here verify our theory and compare with related algorithms on both synthetic data and large-scale machine learning applications.

**Synthetic data** We consider a standard Gaussian model where $x_i \sim \mathcal{N}(\theta, 1), \theta \sim \mathcal{N}(0, 1)$. 1000 data samples $\{x_i\}$ are generated, and every minibatch in the stochastic gradient is of size 10. The test function is defined as $\phi(\theta) \triangleq \theta^2$, with explicit expression for the posterior average. To evaluate the expectations in the bias and MSE, we average over 200 runs with random initializations.

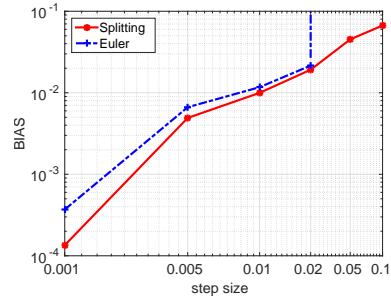

First we compare the invariant measures (with $L = 10^6$) of the proposed splitting integrator and Euler integrator for the SGHMC. Results of the SGLD are omitted since they are not as competitive. Figure 1 plots the biases with different step sizes. It is clear that the Euler integrator has larger biases

Figure 1: Comparisons of symmetric splitting and Euler integrators.

in the invariant measure, and quickly explodes when the step size becomes large, which does not happen for the splitting integrator. In real applications we also find this happen frequently (shown in the next section), making the Euler scheme an unstable integrator.

Next we examine the asymptotically optimal step size rates for the SGHMC. From the theory we know these are $\alpha = 1/3$ for the bias and $\alpha = 1/5$ for the MSE, in both fixed-step-size SGHMC (SGHMC-F) and decreasing-step-size SGHMC (SGHMC-D). For the step sizes, we did a grid search to select the best prefactors, which resulted in $h = 0.033 \times L^{-\alpha}$ for the SGHMC-F and $h_l = 0.045 \times l^{-\alpha}$ for the SGHMC-D, with different $\alpha$ values. We plot the traces of the bias for the SGHMC-D and the MSE for the SGHMC-F in Figure 2. Similar results for the bias of the SGHMC-F and the MSE of the SGHMC-D are plotted in Appendix K. We find that when rates are smaller than the theoretically optimal rates, *i.e.*, $\alpha = 1/3$ (bias) and $\alpha = 1/5$ (MSE), the bias and MSE tend to decrease faster than the optimal rates at the beginning (especially for the SGHMC-F), but eventually they slow down and are surpassed by the optimal rates, consistent with the *asymptotic* theory. This also suggests that if only a small number of iterations were feasible, setting a larger step size than the theoretically optimal one might be beneficial in practice.

Finally, we study the relative convergence speed of the SGHMC and SGLD. We test both fixed-step-size and decreasing-step-size versions. For fixed-step-size experiments, the step sizes are set to $h = CL^{-\alpha}$, with $\alpha$ chosen according to the theory for SGLD and SGHMC. To provide a fair comparison, the constants $C$ are selected via a grid search from $10^{-3}$ to 0.5 with an interval of 0.002 for $L = 500$, it is then fixed in the other runs with different $L$ values. The parameter $D$ in the SGHMC is selected within $(10, 20, 30)$ as well. For decreasing-step-size experiments, an initial step size is chosen within $[0.003, 0.05]$ with an interval of 0.002 for different algorithms[‡‡], and then it decreases according to their theoretical optimal rates. Figure 3 shows a comparison of the biases for the SGHMC and SGLD. As indicated by both theory and experiments, the SGHMC with the splitting integrator yields a faster convergence speed than the SGLD with an Euler integrator.

**Large-scale machine learning applications** For real applications, we test the SGLD with an Euler integrator, the SGHMC with the splitting integrator (SGHMC-S), and the SGHMC with an

---

[‡‡]Using the same initial step size is not fair because the SGLD requires much smaller step sizes.

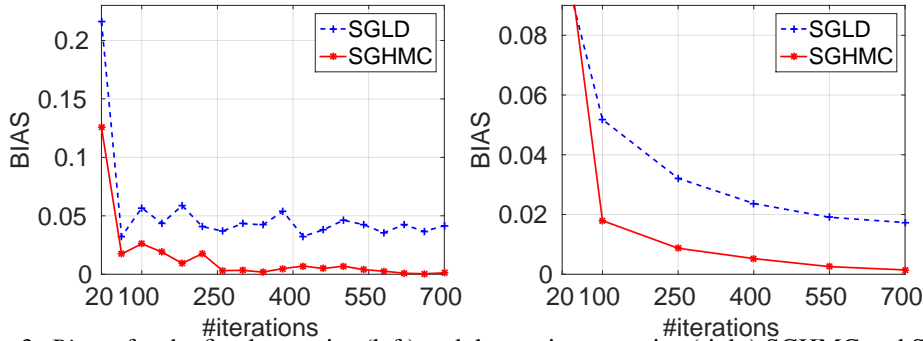

Figure 3: *Biases* for the fixed-step-size (left) and decreasing-step-size (right) SGHMC and SGLD.

Euler integrator (SGHMC-E). First we test them on the latent Dirichlet allocation model (LDA) [16]. The data used consists of 10M randomly downloaded documents from *Wikipedia*, using scripts provided in [17]. We randomly select 1K documents for testing and validation, respectively. As in [17, 18], the vocabulary size is 7,702. We use the *Expanded-Natural* reparametrization trick to sample from the probabilistic simplex [19]. The step sizes are chosen from $\{2, 5, 8, 20, 50, 80\} \times 10^{-5}$, and parameter $D$ from $\{20, 40, 80\}$. The minibatch size is set to 100, with one pass of the whole data in the experiments (and therefore $L = 100K$). We collect 300 posterior samples to calculate test perplexities, with a standard holdout technique as described in [18].

Next a recently studied sigmoid belief network model (SBN) [20] is tested, which is a directed counterpart of the popular RBM model. We use a one layer model where the bottom layer corresponds to binary observed data, which is generated from the hidden layer (also binary) via a sigmoid function. As shown in [18], the SBN is readily learned by SG-MCMCs. We test the model on the MNIST dataset, which consists of 60K hand written digits of size $28 \times 28$ for training, and 10K for testing. Again the step sizes are chosen from $\{3, 4, 5, 6\} \times 10^{-4}$, $D$ from $\{0.9, 1, 5\}/\sqrt{h}$. The minibatch is set to 200, with 5000 iterations for training. Like applied for the RBM [21], an advance technique called anneal importance sampler (AIS) is adopted for calculating test likelihoods.

We briefly describe the results here, more details are provided in Appendix K. For LDA with 200 topics, the best test perplexities for the SGHMC-S, SGHMC-E and SGLD are 1168, 1180 and 2496, respectively; while these are 1157, 1187 and 2511, respectively, for 500 topics. Similar to the synthetic experiments, we also observed SGHMC-E crashed when using large step sizes. This is illustrated more clearly in Figure 4. For the SBN with 100 hidden units, we obtain negative test log-likelihoods of 103, 105 and 126 for the SGHMC-S, SGHMC-E and SGLD, respectively; and these are 98, 100, and 110 for 200 hidden units. Note the SGHMC-S on SBN yields state-of-the-art results on test likelihoods compared to [22], which was 113 for 200 hidden units. A decrease of 2 units in the neg-log-likelihood with AIS is considered to be a reasonable gain [20], which is approximately equal to the gain from a shallow to a deep model [22]. SGHMC-S is more accuracy and robust than SGHMC-E due to its 2nd-order splitting integrator.

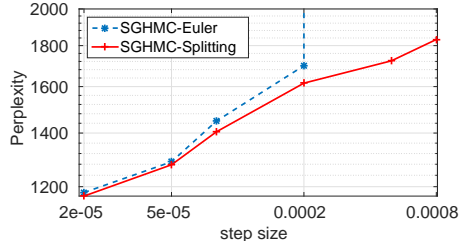

Figure 4: SGHMC with 200 topics. The Euler explodes with large step sizes.

## 6 Conclusion

For the first time, we develop theory to analyze finite-time ergodic errors, as well as asymptotic invariant measures, of general SG-MCMCs with high-order integrators. Our theory applies for both fixed and decreasing step size SG-MCMCs, which are shown to be equivalent in terms of convergence rates, and are faster with our proposed 2nd-order integrator than previous SG-MCMCs with 1st-order Euler integrators. Experiments on both synthetic and large real datasets validate our theory. The theory also indicates that with increasing order of numerical integrators, the convergence rate of an SG-MCMC is able to theoretically approach the standard MCMC convergence rate. Given the theoretical convergence results, SG-MCMCs can be used effectively in real applications.

**Acknowledgments** Supported in part by ARO, DARPA, DOE, NGA and ONR. We acknowledge Jonathan C. Mattingly and Chunyuan Li for inspiring discussions; David Carlson for the AIS codes.

## Footnotes

*For completeness, we provide mean sample-path properties of the SGHMC (similar to [7]) in Appendix J.

†More details of the equation are provided in Appendix B. Specifically, under mild conditions on $F$, we can expand the operator $e^{h\mathcal{L}}$ up to the $m$th-order ($m \geq 1$) such that the remainder terms are bounded by $O(h^{m+1})$. Refer to [10] for more details. We will assume these conditions to hold for the $F$'s in this paper.

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
