[Supplementary Material]

# Supplementary Material for:
# On the Convergence of Stochastic Gradient MCMC Algorithms with High-Order Integrators

Changyou Chen[†]    Nan Ding[‡]    Lawrence Carin[†]
[†]Dept. of Electrical and Computer Engineering, Duke University, Durham, NC, USA
[‡]Google Inc., Venice, CA, USA
cchangyou@gmail.com; dingnan@google.com; lcarin@duke.edu

## A    Representative Stochastic Gradient MCMC Algorithms

This section briefly introduces three recently proposed stochastic gradient MCMC algorithms, including the stochastic gradient Langevin dynamic (SGLD) [4], the stochastic gradient Hamiltonian MCMC (SGHMC) [1], and the stochastic gradient Nosé-Hoover thermostat [2] (SGNHT).

Given data $\mathbf{X} = \{\mathbf{x}_1, \cdots, \mathbf{x}_N\}$, a generative model $p(\mathbf{X} \,|\, \boldsymbol{\theta}) = \prod_{i=1}^{N} p(\mathbf{x}_i \,|\, \boldsymbol{\theta})$ with model parameter $\boldsymbol{\theta}$, and prior $p(\boldsymbol{\theta})$, we want to compute the posterior:

$$\pi(\boldsymbol{\theta}) \triangleq p(\boldsymbol{\theta} \,|\, \mathbf{X}) \propto p(\mathbf{X} \,|\, \boldsymbol{\theta})p(\boldsymbol{\theta}) \triangleq e^{-U(\boldsymbol{\theta})} \ .$$

### A.1    Stochastic gradient Langevin dynamics

The SGLD [4] is based on the following 1st-order Langevin dynamic defined as:

$$\mathrm{d}\boldsymbol{\theta} = -\frac{1}{2}\nabla_{\boldsymbol{\theta}} U(\boldsymbol{\theta})\mathrm{d}t + \mathrm{d}W \ , \tag{10}$$

where $W$ is the standard Brownian motion. We can show via the Fokker–Planck equation that the equilibrium distribution of (10) is:

$$p(\boldsymbol{\theta}) = \pi(\boldsymbol{\theta}) \ .$$

As described in the main text, when sampling from this continuous-time diffusion, two approximations are adopted, *e.g.*, a numerical integrator and a stochastic gradient version $\tilde{U}_l(\boldsymbol{\theta}_{(l-1)h})$ of the log-likelihood $U(\boldsymbol{\theta})$ from the $l$-th minibatch. This results in the following SGLD algorithm.

---

**Algorithm 1:** Stochastic Gradient Langevin Dynamics

---

**Input**: Parameters $h$.
Initialize $\boldsymbol{\theta}_0 \in \mathbf{R}^n$ ;
**for** $l = 1, 2, \ldots$ **do**
  Evaluate $\nabla \tilde{U}_l(\boldsymbol{\theta}_{(l-1)h})$ from the $l$-th minibatch ;
  $\boldsymbol{\theta}_{lh} = \boldsymbol{\theta}_{(l-1)h} - \nabla \tilde{U}_l(\boldsymbol{\theta}_{(l-1)h})h + \sqrt{2h}\,\mathcal{N}(0, I)$;
**end**

---

### A.2    Stochastic gradient Hamiltonian MCMCs

The SGHMC [1] is based on the 2nd-order Langevin dynamic defined as:

$$\begin{cases} \mathrm{d}\boldsymbol{\theta} & = \mathbf{p}\,\mathrm{d}t \\ \mathrm{d}\,\mathbf{p} & = -\nabla_{\boldsymbol{\theta}} U(\boldsymbol{\theta})\mathrm{d}t - D\,\mathbf{p}\,\mathrm{d}t + \sqrt{2D}\mathrm{d}W \ , \end{cases} \tag{11}$$

where $D$ is a constant independent of $\boldsymbol{\theta}$ and $\mathbf{p}$. Again we can show that the equilibrium distribution of (11) is:

$$P(\boldsymbol{\theta}, \mathbf{p}) \propto e^{-U(\boldsymbol{\theta}) + \frac{\mathbf{p}^T \mathbf{p}}{2}} \ .$$

Similar to the SGLD, we use the Euler scheme to simulate the dynamic (11), shown in Algorithm 2.

---
**Algorithm 2:** Stochastic Gradient Hamiltonian MCMC
---
**Input**: Parameters $h, D$.
Initialize $\boldsymbol{\theta}_0 \in \mathbf{R}^n$, $\mathbf{p}_0 \sim \mathcal{N}(0, \mathbf{I})$ ;
**for** $l = 1, 2, \ldots$ **do**
    Evaluate $\nabla \tilde{U}_l(\boldsymbol{\theta}_{(l-1)h})$ from the $l$-th minibatch ;
    $\mathbf{p}_{lh} = \mathbf{p}_{(l-1)h} - D\,\mathbf{p}_{(l-1)h}\,h - \nabla \tilde{U}_l(\boldsymbol{\theta}_{(l-1)h})h + \sqrt{2Dh}\,\mathcal{N}(0,1)$;
    $\boldsymbol{\theta}_{lh} = \boldsymbol{\theta}_{(l-1)h} + \mathbf{p}_{lh}\,h$;
**end**
---

## A.3 Stochastic gradient Nośe-Hoover thermostats

The SGNHT [2] is based on the Nośe-Hoover thermostat defined as:

$$
\begin{cases}
\mathrm{d}\boldsymbol{\theta} & = \mathbf{p}\,\mathrm{d}t \\
\mathrm{d}\,\mathbf{p} & = -\nabla_{\boldsymbol{\theta}} U(\boldsymbol{\theta})\mathrm{d}t - \xi\,\mathbf{p}\,\mathrm{d}t + \sqrt{2D}\mathrm{d}W \\
\mathrm{d}\xi & = \left(\mathbf{p}^T\,\mathbf{p}\,/n - 1\right) ,
\end{cases} \tag{12}
$$

If $D$ is independent of $\boldsymbol{\theta}$ and $\mathbf{p}$, it can also be shown that the equilibrium distribution of (12) is [2]:

$$
P(\boldsymbol{\theta}, \mathbf{p}, \xi) \propto e^{-U(\boldsymbol{\theta}) - \frac{1}{2}\mathbf{p}^T\mathbf{p} + \frac{1}{2}(\xi - D)^2} .
$$

The SGNHT is much more interesting than the SGHMC when considering subsampling data in each iteration, as the covariance $D$ in SGHMC is hard to estimate, a thermostat is used to adaptively control the system temperature, thus automatically estimate the unknown $D$. The whole algorithm is shown in Algorithm 3.

---
**Algorithm 3:** Stochastic Gradient Nosé-Hoover Thermostats
---
**Input**: Parameters $h, D$.
Initialize $\boldsymbol{\theta}_0 \in \mathbf{R}^n$, $\mathbf{p}_0 \sim \mathcal{N}(0, \mathbf{I})$, and $\xi_0 = D$ ;
**for** $l = 1, 2, \ldots$ **do**
    Evaluate $\nabla \tilde{U}_l(\boldsymbol{\theta}_{(l-1)h})$ from the $l$-th minibatch ;
    $\mathbf{p}_{lh} = \mathbf{p}_{(l-1)h} - \xi_{(l-1)h}\,\mathbf{p}_{(l-1)h}\,h - \nabla \tilde{U}_l(\boldsymbol{\theta}_{(l-1)h})h + \sqrt{2Dh}\,\mathcal{N}(0,I)$;
    $\boldsymbol{\theta}_{lh} = \boldsymbol{\theta}_{(l-1)h} + \mathbf{p}_{lh}\,h$;
    $\xi_{lh} = \xi_{(l-1)h} + (\frac{1}{n}\mathbf{p}_{lh}^{\top}\mathbf{p}_{lh} - 1)h$;
**end**
---

# B More Details on Kolmogorov's Backward Equation

The generator $\mathcal{L}$ is used in the formulation of Kolmogorov's backward equation, which intuitively tells us how the expected value of any suitably smooth statistic of $\mathbf{X}$ evolves in time. More precisely:

**Definition 11** (Kolmogorov's Backward Equation). *Let* $u(t, \mathbf{x}) = \mathbb{E}\left[\phi(\mathbf{X}_t)\right]$, *then* $u(t, \mathbf{x})$ *satisfies the following partial differential equation, known as* Kolmogorov's backward equation*:*

$$
\begin{cases}
\frac{\partial u}{\partial t}(t, \mathbf{x}) = \mathcal{L}u(t, \mathbf{x}) , & t > 0, \mathbf{x} \in \mathbb{R}^n \\
u(0, \mathbf{x}) = \phi(\mathbf{x}), & \mathbf{x} \in \mathbf{R}^n
\end{cases} \tag{13}
$$

Based on the definition, we can write $u(t, \cdot) = P_t\phi(\cdot)$ so that $(P_t)_{t \geq 0}$ is the transition semigroup associated with the Markov process $(\mathbf{X}(t, \mathbf{x}))_{t \geq 0, \mathbf{x} \in \mathbb{R}^n}$ [23] (also called the Kolmogorov operator). Note that the *Kolmogorov's backward equation* can be written in another form as:

$$
u(t, \mathbf{x}) = \mathbb{E}\left[\phi(\mathbf{X}_t)\right] = e^{t\mathcal{L}}\phi(\mathbf{x}) , \tag{14}
$$

where $e^{t\mathcal{L}}$ is the exponential map operator associated with the generator defined as:

$$
e^{t\mathcal{L}} \triangleq \mathbb{I} + \sum_{i=1}^{\infty} \frac{(t\mathcal{L})^i}{i!} ,
$$

with $\mathbb{I}$ being the identity map. This is obtained by expanding $u(t, \mathbf{x})$ in time by using Taylor expansion [23]:

$$
\begin{aligned}
u(t, \mathbf{x}) &= u(0, \mathbf{x}) + \sum_{i=1}^{\infty} \frac{t^i}{i!} \frac{\mathrm{d}^i}{\mathrm{d}t^i} u(t, \mathbf{x})\big|_{t=0} \\
&= u(0, \mathbf{x}) + \sum_{i=1}^{\infty} \frac{t^i}{i!} \frac{\mathrm{d}^{i-1}}{\mathrm{d}t^{i-1}} \frac{\mathrm{d}}{\mathrm{d}t} u(t, \mathbf{x})\big|_{t=0} \\
&= u(0, \mathbf{x}) + \sum_{i=1}^{\infty} \frac{t^i}{i!} \mathcal{L} \frac{\mathrm{d}^{i-1}}{\mathrm{d}t^{i-1}} u(t, \mathbf{x})\big|_{t=0} \\
&= \phi(\mathbf{x}) + \sum_{i=1}^{\infty} \frac{t^i}{i!} \mathcal{L}^i \phi(\mathbf{x}) = e^{t\mathcal{L}} \phi(\mathbf{x}) \, .
\end{aligned}
\tag{15}
$$

The form (14) instead of the original form (13) of the Kolmogorov's backward equation is used in our analysis. To be able to expand the form (14) to some particular order such that remainder terms are bounded, the following assumption is required [24].

**Assumption 3.** *Assume 1) $F(\mathbf{X})$ is $C^{\infty}$ with bounded derivatives of any order, furthermore, and 2) $|F(\mathbf{x})| \leq C(1+|\mathbf{X}|^s)$ for some positive integer $s$. Under these assumptions, series of the generator expansion can be bounded, thus (15) can be written in the following form [6, 24]:*

$$
u(t, \mathbf{x}) = \phi(\mathbf{x}) + \sum_{i=1}^{\ell} \frac{t^i}{i!} \mathcal{L}^i \phi(\mathbf{x}) + t^{\ell+1} r_\ell(F, \phi)(\mathbf{x}) \, ,
\tag{16}
$$

*with $|r_\ell(F, \phi)(\mathbf{x})| \leq C_\ell(1 + |\mathbf{x}|^{k_\ell})$ for some constant $C_\ell, k_\ell$.*

## C   More Comments on Assumption 1

Assumption 1 assumes that the solution functional $\psi$ of the Poisson equation (4) satisfies: $\psi$ and its up to 3-rd order derivatives, $\mathcal{D}^k \psi$, are bounded by a function $\mathcal{V}$, *i.e.*, $\|\mathcal{D}^k \psi\| \leq C_k \mathcal{V}^{p_k}$ for $k = (0, 1, 2, 3)$, $C_k, p_k > 0$. Furthermore, $\mathcal{V}$ is smooth such that $\sup_{s \in (0,1)} \mathcal{V}^p(s\mathbf{X} + (1-s)\mathbf{Y}) \leq C(\mathcal{V}^p(\mathbf{X}) + \mathcal{V}^p(\mathbf{Y})), \forall \mathbf{X}, \mathbf{Y}, p \leq p^* \triangleq \max\{2p_k\}$ for some $C > 0$. Finally, $\sup_l \mathbb{E}\mathcal{V}^p(\mathbf{X}_{lh}) < \infty$ for $p \leq p^*$. This is summarized as:

$$
\sup_l \mathbb{E}\mathcal{V}^p(\mathbf{X}_{lh}) < \infty
\tag{17}
$$

$$
\sup_{s \in (0,1)} \mathcal{V}^p(s\mathbf{X} + (1-s)\mathbf{Y}) \leq C(\mathcal{V}^p(\mathbf{X}) + \mathcal{V}^p(\mathbf{Y}))
\tag{18}
$$

$$
\|\mathcal{D}^k \psi\| \leq C_k \mathcal{V}^{p_k}
\tag{19}
$$

Compared to the SGLD case [6], in our proofs, we only need $k$ be up to 3 in (19) instead of 4. More specifically, the proof for the bias only needs $k$ be up to 0 given other assumptions in this paper, and the proof for the MSE needs $k$ be up to 3.

As long as the corresponding SDE is hypoelliptic, meaning that the Brownian motion $W$ is able to propagate to the other variables of the dynamics [12], *e.g.*, the model parameter $\boldsymbol{\theta}$ in SGHMC, we can extend Assumption 4.1 of [6] to our setting. Thus we have that (17) is equivalent to finding a function $\mathcal{V} : \mathbf{R}^n \to [1, \infty]$ ($n$ is the dimension of $\mathbf{x}$, *e.g.*, including the momentum in SGHMC), which tends to infinity as $\mathbf{x} \to \infty$, and is twice differentiable with bounded second derivatives and satisfies the following conditions:

1. $\mathcal{V}$ is a Lyapunov function of the SDE, *i.e.*, there exists constants $\alpha, \beta > 0$, such that for $\mathbf{x} \in \mathbf{R}^n$, we have $\langle \nabla_{\mathbf{x}} \mathcal{V}(\mathbf{x}), F(\mathbf{x}) \rangle \leq -\alpha \mathcal{V}(\mathbf{x}) + \beta$.

2. There exists an exponent $p_H \geq 2$ such that $\mathbb{E} \left\| \tilde{F}(\mathbf{x}) - \mathbb{E}_s \tilde{F}(\mathbf{x}) \right\| \lesssim \mathcal{V}^{p_H}(\mathbf{x})$, where $\mathbb{E}_s$ means expectation with respect to the random permutation of the data, $\mathbb{E}$ means expectation with respect to the randomness of the dynamic with Brownian motion. Furthermore, for $\mathbf{x} \in \mathbf{R}^n$, we have: $\|\nabla_{\mathbf{x}} \mathcal{V}(\mathbf{x})\|^2 + \|F(\mathbf{x})\|^2 \lesssim \mathcal{V}(\mathbf{x})$.

Similar to [6], (18) is an extra condition that needs to be satisfied, and (19) is more subtle and needs more assumptions to verify in this case. We will not address these issues because it is out of the scope of the paper.

## D   The Proof of Theorem 2

*Proof.* For an SG-MCMC with a $K$th-order integrator, according to Definition 1 and (3), we have:

$$\mathbb{E}[\psi(\mathbf{X}_{lh})] = \tilde{P}_h^l \psi(\mathbf{X}_{(l-1)h}) = e^{h\tilde{\mathcal{L}}_l}\psi(\mathbf{X}_{(l-1)h}) + O(h^{K+1})$$

$$= \left(\mathbb{I} + h\tilde{\mathcal{L}}_l\right)\psi(\mathbf{X}_{(l-1)h}) + \sum_{k=2}^{K}\frac{h^k}{k!}\tilde{\mathcal{L}}_l^k\psi(\mathbf{X}_{(l-1)h}) + O(h^{K+1}), \qquad (20)$$

where $\mathbb{I}$ is the identity map. Sum over $l = 1, \cdots, L$ in (20), take expectation on both sides, and use the relation $\tilde{\mathcal{L}}_l = \mathcal{L} + \Delta V_l$ to expand the first order term. We obtain

$$\sum_{l=1}^{L}\mathbb{E}[\psi(\mathbf{X}_{lh})] = \psi(\mathbf{X}_0) + \sum_{l=1}^{L-1}\mathbb{E}[\psi(\mathbf{X}_{lh})] + h\sum_{l=1}^{L}\mathbb{E}[\mathcal{L}\psi(\mathbf{X}_{(l-1)h})]$$

$$+ h\sum_{l=1}^{L}\mathbb{E}[\Delta V_l\psi(\mathbf{X}_{(l-1)h})] + \sum_{k=2}^{K}\frac{h^k}{k!}\sum_{l=1}^{L}\mathbb{E}[\tilde{\mathcal{L}}_l^k\psi(\mathbf{X}_{(l-1)h})] + O(Lh^{K+1}).$$

Divide both sides by $Lh$, use the Poisson equation (4), and reorganize terms. We have:

$$\mathbb{E}[\frac{1}{L}\sum_l\phi(\mathbf{X}_{lh}) - \bar{\phi}] = \frac{1}{L}\sum_{l=1}^{L}\mathbb{E}[\mathcal{L}\psi(\mathbf{X}_{(l-1)h})] \qquad (21)$$

$$= \frac{1}{Lh}\left(\mathbb{E}[\psi(\mathbf{X}_{lh})] - \psi(\mathbf{X}_0)\right) - \frac{1}{L}\sum_l\mathbb{E}[\Delta V_l\psi(\mathbf{X}_{(l-1)h})] - \sum_{k=2}^{K}\frac{h^{k-1}}{k!L}\sum_{l=1}^{L}\mathbb{E}[\tilde{\mathcal{L}}_l^k\psi(\mathbf{X}_{(l-1)h})] + O(h^K)$$

To transform terms containing $\tilde{\mathcal{L}}_l^k\,(k \geq 2)$ to high-order terms, based on ideas from [12], we apply the following procedure. First replace $\psi$ with $\tilde{\mathcal{L}}_l^{K-1}\psi$ from (20) to (21), and apply the same logic for $\tilde{\mathcal{L}}_l^{K-1}\psi$ as for $\psi$ in the above derivations, but this time expand in (20) up to the order of $O(h^2)$, instead of the previous order $O(h^{K+1})$. After simplification, we obtain:

$$\sum_l\mathbb{E}[\tilde{\mathcal{L}}_l^K\psi(\mathbf{X}_{(l-1)h})] = O\left(\frac{1}{h} + Lh\right) \qquad (22)$$

Similarly, replace $\psi$ with $\tilde{\mathcal{L}}_l^{K-2}\psi$ from (20) to (21), follow the same derivations as for $\tilde{\mathcal{L}}_l^{K-1}\psi$, but expand in (20) up to the order of $O(h^3)$ instead of $O(h^2)$. We have:

$$\sum_l\mathbb{E}[\tilde{\mathcal{L}}_l^{K-1}\psi(\mathbf{X}_{(l-1)h})] = O\left(\frac{1}{h} + Lh^2\right) + \frac{h}{2}\sum_{l=1}^{L}\mathbb{E}[\tilde{\mathcal{L}}_l^K\psi(\mathbf{X}_{(l-1)h})] = O\left(\frac{1}{h} + Lh^2\right),$$
$$\tag{23}$$

where the last equation in (23) is obtained by substituting (22) into it and collecting low order terms. By induction on $k$, it is easy to show that for $2 \leq k \leq K$, we have:

$$\sum_l\mathbb{E}[\tilde{\mathcal{L}}_l^k\psi(\mathbf{X}_{(l-1)h})] = O\left(\frac{1}{h} + Lh^{K-k+1}\right), \qquad (24)$$

Substituting (24) into (21), after simplification, we have: $\mathbb{E}\left(\frac{1}{L}\sum_l\phi(\mathbf{X}_{lh}) - \bar{\phi}\right)$

$$= \frac{1}{Lh}\underbrace{\left(\mathbb{E}[\psi(\mathbf{X}_{lh})] - \psi(\mathbf{X}_0)\right)}_{C_1} - \frac{1}{L}\sum_l\mathbb{E}[\Delta V_l\psi(\mathbf{X}_{(l-1)h})] - \sum_{k=2}^{K}O\left(\frac{h^{k-1}}{Lh} + h^K\right) + C_3 h^K,$$

for some $C_3 \geq 0$. According to the assumption, the term $C_1$ is bounded. As a result, collecting low order terms, the bias can be expressed as:

$$\left| \mathbb{E}\hat{\phi} - \bar{\phi} \right| = \left| \mathbb{E}\left( \frac{1}{L}\sum_l \phi(\mathbf{X}_{lh}) - \bar{\phi} \right) \right| = \left| \frac{C_1}{Lh} - \frac{\sum_l \mathbb{E}\Delta V_l \psi(\mathbf{X}_{(l-1)h})}{L} + C_3 h^K \right|$$

$$\leq \left| \frac{C_1}{Lh} \right| + \left| \frac{\sum_l \mathbb{E}\Delta V_l \psi(\mathbf{X}_{(l-1)h})}{L} \right| + \left| C_3 h^K \right| = O\left( \frac{1}{Lh} + \frac{\sum_l \|\mathbb{E}\Delta V_l\|}{L} + h^K \right) ,$$

where the last equation follows from the finiteness assumption of $\psi$, $\|\cdot\|$ denotes the operator norm and is bounded in the space of $\psi$ due to the assumptions. This completes the proof. $\square$

## E  The Proof of Theorem 3

*Proof.* For a $K$-order integrator, from Theorem 2, we can expand $\mathbb{E}(\psi(\mathbf{X}_{lh}))$ as:

$$\mathbb{E}(\psi(\mathbf{X}_{lh})) = (\mathbb{I} + h(\mathcal{L} + \Delta V_l))\psi(\mathbf{X}_{(l-1)h}) + \sum_{k=2}^{K} \frac{h^k}{k!}\tilde{\mathcal{L}}_l^k \psi(\mathbf{X}_{(l-1)h}) + O(h^{K+1}) .$$

Sum over $l$ from 1 to $L + 1$ and simplify, we have:

$$\sum_{l=1}^{L} \mathbb{E}(\psi(\mathbf{X}_{lh})) = \sum_{l=1}^{L} \psi(\mathbf{X}_{(l-1)h}) + h\sum_{l=1}^{L} \mathcal{L}\psi(\mathbf{X}_{(l-1)h}) + h\sum_{l=1}^{L} \Delta V_l \psi(\mathbf{X}_{(l-1)h})$$

$$+ \sum_{k=2}^{K} \frac{h^k}{k!} \sum_{l=1}^{L} \tilde{\mathcal{L}}_l^k \psi(\mathbf{X}_{(l-1)h}) + O(Lh^{K+1}) .$$

Substitute the Poisson equation (4) into the above equation, divide both sides by $Lh$ and rearrange related terms, we have

$$\hat{\phi} - \bar{\phi} = \frac{1}{Lh}(\mathbb{E}\psi(\mathbf{X}_{Lh}) - \psi(\mathbf{X}_0)) - \frac{1}{Lh}\sum_{l=1}^{L} \left( \mathbb{E}\psi(\mathbf{X}_{(l-1)h}) - \psi(\mathbf{X}_{(l-1)h}) \right)$$

$$- \frac{1}{L}\sum_{l=1}^{L} \Delta V_l \psi(\mathbf{X}_{(l-1)h}) - \sum_{k=2}^{K} \frac{h^{k-1}}{2L} \sum_{l=1}^{L} \tilde{\mathcal{L}}_l^k \psi(\mathbf{X}_{(l-1)h}) + O(h^K)$$

Taking square and expectation on both sides, since the terms $\left( \mathbb{E}\psi(\mathbf{X}_{(l-1)h}) - \psi(\mathbf{X}_{(l-1)h}) \right)$ and $\Delta V_l \psi(\mathbf{X}_{(l-1)h})$ are martingale, it is then easy to see there exists some positive constant $C$, such that

$$\mathbb{E}\left( \hat{\phi} - \bar{\phi} \right)^2 \leq C\mathbb{E}\left( \underbrace{\frac{(\mathbb{E}\psi(\mathbf{X}_{Lh}) - \psi(\mathbf{X}_0))^2}{L^2 h^2}}_{A_1} + \underbrace{\frac{1}{L^2 h^2}\sum_{l=1}^{L} \left( \mathbb{E}\psi(\mathbf{X}_{(l-1)h}) - \psi(\mathbf{X}_{(l-1)h}) \right)^2}_{A_2} \right.$$

$$\left. + \frac{1}{L^2}\sum_{l=1}^{L} \Delta V_l^2 \psi(\mathbf{X}_{(l-1)h}) + \underbrace{\sum_{k=2}^{K} \frac{h^{2(k-1)}}{k! L^2} \left( \sum_{l=1}^{L} \tilde{\mathcal{L}}_l^k \psi(\mathbf{X}_{(l-1)h}) \right)^2}_{A_3} + h^{2K} \right) \quad (25)$$

$A_1$ is easily bounded by the assumption that $\|\psi\| \leq V^{p_0} < \infty$, the expectation of $A_3$ can also be shown to be bounded later in (31). Now we show that $A_2$ is bounded as well by deriving the following bound: $\mathbb{E}(\psi(\mathbf{X}_{lh})) - \psi(\mathbf{X}_{lh}) \leq C_1\sqrt{h} + O(h)$ for $C_1 \geq 0$. To do this, it is enough to consider the 2nd order symmetric splitting scheme, as higher order integrators generally introduce higher order errors. Furthermore, we see that different splitting schemes, *e.g.*, ABOBA and OABAO, are essentially equivalent as long as they are symmetric [15], thus we focus on the ABOAB scheme in the proof. Let the flow propagators (mappings) of 'A' , 'B' and 'O' be denoted as $\tilde{\varphi}_h^A$, $\tilde{\varphi}_h^B$ and

$\tilde{\varphi}_h^{O_l}$ respectively. Since $\tilde{\varphi}_h^A$ and $\tilde{\varphi}_h^B$ are deterministic, we combine them and use $\tilde{\varphi}_h^{AB}$ to represent the composition flow $\tilde{\varphi}_h^A \circ \tilde{\varphi}_h^B$. We further decompose $\tilde{\varphi}_h^{O_l}$ into the deterministic part $\tilde{\varphi}_h^O$ and the stochastic part $\tilde{\varphi}_h^{\zeta}$ from the brownian motion, then in the iteration for the current minibatch, we can express the flow evolution as:

$$\mathbf{X}_{lh} = \tilde{\varphi}_h^{AB} \circ \left( \tilde{\varphi}_h^O \circ \tilde{\varphi}_h^\zeta \right) \circ \tilde{\varphi}_h^{AB}(\mathbf{X}_{(l-1)h})$$
$$= \tilde{\varphi}_h^{AB} \left( \tilde{\varphi}_h^O \left( \tilde{\varphi}_h^{AB}(\mathbf{X}_{(l-1)h}) \right) + \sqrt{2Dh}\zeta_l \right) , \qquad (26)$$

where $\zeta_l$ is a $n$-dimensional independent Gaussian random variables.

From Assumption 1 we know that both $\tilde{\varphi}_h^O$ and $\tilde{\varphi}_h^{AB}$ have bounded derivatives. To simplify the representation, we denote $\tilde{\mathbf{X}}_l \triangleq \tilde{\varphi}_h^{AB} \left( \tilde{\varphi}_h^O \left( \tilde{\varphi}_h^{AB}(\mathbf{X}_{(l-1)h}) \right) \right)$. Now we can expanded $\mathbf{X}_{lh}$ from (26) using Taylor expansion as:

$$\mathbf{X}_{lh} = \tilde{\varphi}_h^{AB} \left( \tilde{\varphi}_h^O \left( \tilde{\varphi}_h^{AB}(\mathbf{X}_{(l-1)h}) \right) + \sqrt{2Dh}\zeta_l \right)$$
$$= \tilde{\mathbf{X}}_l + \mathcal{D}\tilde{\mathbf{X}}_l \left[ \sqrt{2Dh}\zeta_l \right] + \frac{1}{2}\mathcal{D}^2\tilde{\mathbf{X}}_l \left[ \sqrt{2Dh}\zeta_l, \sqrt{2Dh}\zeta_l \right] + O(h\zeta_l^2) \qquad (27)$$

Using the relation (27), for the solution $\psi$ of the Poisson equation (4) applied on $\mathbf{X}_{lh}$, we can bow expand it up to 3 orders from the Taylor theory:

$$\psi(\mathbf{X}_{lh}) = \psi\left( \tilde{\varphi}_h^{AB} \left( \tilde{\varphi}_h^O \left( \tilde{\varphi}_h^{AB}(\mathbf{X}_{(l-1)h}) \right) \right) + \mathcal{D}\tilde{\mathbf{X}}_l \left[ \sqrt{2Dh}\zeta_l \right] + \frac{1}{2}\mathcal{D}^2\tilde{\mathbf{X}}_l \left[ \sqrt{2Dh}\zeta_l, \sqrt{2Dh}\zeta_l \right] + O(h\zeta_l^2) \right)$$

$$= \psi\left( \tilde{\mathbf{X}}_l \right) + \underbrace{\mathcal{D}\psi(\tilde{\mathbf{X}}_l) \left[ \mathcal{D}\tilde{\mathbf{X}}_l \left[ \sqrt{2Dh}\zeta_l \right] \right]}_{M_1} + \underbrace{\frac{1}{2}\mathcal{D}\psi(\tilde{\mathbf{X}}_l) \left[ \mathcal{D}^2\tilde{\mathbf{X}}_l \left[ \sqrt{2Dh}\zeta_l, \sqrt{2Dh}\zeta_l \right] \right]}_{S_1}$$

$$+ \underbrace{\frac{1}{2}\mathcal{D}^2\psi(\tilde{\mathbf{X}}_l) \left[ \left( \mathcal{D}\tilde{\mathbf{X}}_l \left[ \sqrt{2Dh}\zeta_l \right] + \frac{1}{2}\mathcal{D}^2\tilde{\mathbf{X}}_l \left[ \sqrt{2Dh}\zeta_l, \sqrt{2Dh}\zeta_i \right] \right)^{2\otimes} \right]}_{S_2} \qquad (28)$$

$$+ \underbrace{\frac{1}{2}\int_0^1 s^2 \mathcal{D}^3\psi(s\mathbf{X}_{(l-1)h} + (1-s)\tilde{\mathbf{X}}_l) \left[ \left( \mathcal{D}\tilde{\mathbf{X}}_l \left[ \sqrt{2Dh}\zeta_l \right] + \frac{1}{2}\mathcal{D}^2\tilde{\mathbf{X}}_l \left[ \sqrt{2Dh}\zeta_l, \sqrt{2Dh}\zeta_l \right] \right)^{3\otimes} \right]}_{R}$$

where $[(\mathbf{X})^N \otimes] \triangleq \underbrace{[\mathbf{X}, \cdots, \mathbf{X}]}_{N}$.

Note that the vector fields inside the brackets in the above expression are all bounded due to Assumption 1. As a result, we can show that $M_1, S_1, S_2$ and $R$ are bounded by the boundedness assumption on $\psi$ and its derivatives. Specifically, in the following we will use $a \lesssim b$ to represent there is a $C \geq 0$ such that $a \leq Cb$. Let $\tilde{\varphi}_{h_l}(\mathbf{x}) \triangleq \tilde{\varphi}_h^{OA} \left( \tilde{\varphi}_h^B \left( \tilde{\varphi}_h^{OA}(\mathbf{X}_{lh} + \mathbf{x}) \right) \right)$, according to the definition of directional derivative, we have

$$\mathcal{D}\tilde{\mathbf{X}}_l \left[ \sqrt{2Dh}\zeta_l \right] \triangleq \lim_{\alpha \to 0} \frac{\tilde{\varphi}_{h_{l-1}}(\alpha\sqrt{2Dh}\zeta_l) - \tilde{\varphi}_{h_{l-1}}(0)}{\alpha}$$
$$= \lim_{\alpha \to 0} \frac{\alpha\sqrt{2Dh}J(0)\zeta_l + O(\alpha)}{\alpha} = \sqrt{2Dh}J(0)\zeta_i ,$$

where $J(\mathbf{x})$ is the Jacobian of $\tilde{\varphi}_{h_{l-1}}(\mathbf{x})$ and is bounded. Thus

$$\mathbb{E}M_1^2 \lesssim h \sup_l \mathbb{E}\mathcal{V}_l^{2p_1} \lesssim h . \qquad (29)$$

Similarly, for $S_1$ and $S_2$, using the assumptions in the theory, we have

$$\mathbb{E}S_1^2 \lesssim h^2 \sup_l \mathbb{E}\mathcal{V}_l^{2p_1} \lesssim h^2$$
$$\mathbb{E}S_2^2 \lesssim (\sqrt{h} + h)^2 \sup_l \mathbb{E}\mathcal{V}_l^{2p_2} \lesssim (\sqrt{h} + h)^2 .$$

For $R$, using Assumption 1, we have

$$\mathbb{E}R^2 \lesssim (\mathbb{E}\mathcal{V}(\mathbf{X}_{(l-1)h})^{2p_3} + \mathbb{E}\mathcal{V}(\tilde{\mathbf{X}}_l)^{2p_3}) \left\| \mathcal{D}\tilde{\varphi}_{h_{l-1}}^{OA}\left[\sqrt{2Dh}\boldsymbol{\zeta}_i\right] + \frac{1}{2}\mathcal{D}^2\tilde{\varphi}_{h_{l-1}}^{OA}\left[\sqrt{2Dh}\boldsymbol{\zeta}_l, \sqrt{2Dh}\boldsymbol{\zeta}_l\right] \right\|^3$$

$$\lesssim h^3 \, .$$

The expectation of $\psi(\mathbf{X}_{lh})$ can be similarly bounded. Collecting low order terms, we have

$$\mathbb{E}\left(\mathbb{E}\left(\psi(\mathbf{X}_{lh})\right) - \psi(\mathbf{X}_{lh})\right)^2 = Ch + O(h^{3/2}) \, ,$$

for some $C > 0$. As a result, the expectation of the $A_2$ term in (25) can be bounded using the above derived bound on $\mathbb{E}\left(\psi(\mathbf{X}_{lh})\right) - \psi(\mathbf{X}_{lh})$.

$$\frac{1}{L^2h^2}\sum_l \mathbb{E}\left(\mathbb{E}\psi(\mathbf{X}_{lh}) - \psi(\mathbf{X}_{lh})\right)^2 = \frac{C}{Lh} + O(\frac{1}{L\sqrt{h}}) \, . \tag{30}$$

Substitute (30) into (25) we can bound the MSE as:

$$\mathbb{E}\left(\hat{\phi} - \bar{\phi}\right)^2$$

$$\lesssim \frac{\frac{1}{L}\sum_l \mathbb{E}\Delta V_l^2\psi(\mathbf{X}_{(l-1)h})}{L} + \sum_{k=2}^K \frac{h^{2(k-1)}}{2L^2}\mathbb{E}\left(\sum_{l=1}^L \tilde{\mathcal{L}}_l^k\psi(\mathbf{X}_{(l-1)h})\right)^2 + \frac{1}{Lh} + \frac{1}{L^2h^2} + O(h^{2K})$$

$$= \frac{\frac{1}{L}\sum_l \mathbb{E}\Delta V_l^2\psi(\mathbf{X}_{(l-1)h})}{L} + \underbrace{\sum_{k=2}^K \frac{h^{2(k-1)}}{2L^2}\left(\sum_{l=1}^L \mathbb{E}\left[\tilde{\mathcal{L}}_l^k\psi(\mathbf{X}_{(l-1)h})\right]\right)^2}_{A_1} + \frac{1}{Lh} + \frac{1}{L^2h^2}$$

$$+ \underbrace{\sum_{k=2}^K \frac{h^{2(k-1)}}{2L^2}\mathbb{E}\left(\sum_{l=1}^L \left(\tilde{\mathcal{L}}_l^k\psi(\mathbf{X}_{(l-1)h}) - \mathbb{E}\tilde{\mathcal{L}}_l^k\psi(\mathbf{X}_{(l-1)h})\right)\right)^2}_{A_2} + O(h^{2K}) \tag{31}$$

$$\leq C\left(\frac{\frac{1}{L}\sum_l \mathbb{E}\|\Delta V_l\|^2}{L} + \frac{1}{Lh} + h^{2K}\right) \tag{32}$$

for some $C > 0$, where (31) follows by using the fact that $\mathbb{E}[\mathbf{X}^2] = \mathbb{E}[(\mathbf{X} - \mathbb{E}\mathbf{X})^2] + (\mathbb{E}\mathbf{X})^2$ for a random variable $\mathbf{X}$. (32) follows by using the bounds in (24) on $A_1$, which is bounded by $O(\frac{1}{L^2h^2} + h^{2K})$. For $A_2$, because the terms $\left(\tilde{\mathcal{L}}_l^k\psi(\mathbf{X}_{(l-1)h}) - \mathbb{E}\tilde{\mathcal{L}}_l^k\psi(\mathbf{X}_{(l-1)h})\right)$ are martingale, we have:

$$A_2 \lesssim \frac{h^{2(k-1)}}{2L^2}\sum_{l=1}^L \mathbb{E}\left(\tilde{\mathcal{L}}_l^k\psi(\mathbf{X}_{(l-1)h}) - \mathbb{E}\tilde{\mathcal{L}}_l^k\psi(\mathbf{X}_{(l-1)h})\right)^2$$

$$\lesssim \frac{1}{Lh}\left(\frac{h^{2k-1}}{L}\sum_{l=1}^L \mathbb{E}(\tilde{\mathcal{L}}_l^k\psi(\mathbf{X}_{(l-1)h}))^2\right) + O\left(\frac{1}{L^2h^2} + h^{2K}\right) = O\left(\frac{1}{Lh} + h^{2K}\right)$$

where we have used (24) and the fact that $\mathbb{E}\tilde{\mathcal{L}}_l^k\psi(\mathbf{X}_{(l-1)h})$ is bounded. Collecting low order terms we get (32). This completes the proof. $\square$

# F  The Proof of Theorem 4

*Proof.* Because the splitting scheme is geometric ergodic, for a test function $\phi$, from the ergodic theorem we have

$$\int_{\mathcal{X}} \phi(\mathbf{x})\tilde{\rho}_h(\mathrm{d}\,\mathbf{x}) = \int_{\mathcal{X}} \mathbb{E}_{\mathbf{x}}\phi(\mathbf{X}_{lh})\tilde{\rho}_h(\mathrm{d}\,\mathbf{x}) \tag{33}$$

for $\forall l \geq 0, \forall \mathbf{x} \in \mathcal{X}$. Average over all the samples $\{\mathbf{X}_{lh}\}$ and let $l$ approach to $\infty$, we have

$$\int_{\mathcal{X}} \phi(\mathbf{x})\tilde{\rho}_h(\mathrm{d}\,\mathbf{x}) = \lim_{L \to \infty} \int_{\mathcal{X}} \frac{1}{L}\sum_{l=1}^{L} \mathbb{E}_{\mathbf{x}}\phi(\mathbf{X}_{lh})\tilde{\rho}_h(\mathrm{d}\,\mathbf{x}) \ .$$

Thus the distance between any invariant measure $\tilde{\rho}_h$ of a high-order integrator and $\rho$ can be bounded as:

$$
\begin{aligned}
d(\tilde{\rho}_h, \rho) &= \sup_{\phi} \left| \int_{\mathcal{X}} \phi(\mathbf{x})\tilde{\rho}_h(\mathrm{d}\,\mathbf{x}) - \int_{\mathcal{X}} \phi(\mathbf{x})\rho(\mathrm{d}\,\mathbf{x}) \right| \\
&= \sup_{\phi} \lim_{L \to \infty} \left| \int_{\mathcal{X}} \left[ \frac{1}{L}\sum_{l=1}^{L} \mathbb{E}_{\mathbf{x}}\phi(\mathbf{X}_{lh}) - \bar{\phi} \right] \tilde{\rho}_h(\mathrm{d}\,\mathbf{x}) \right| \\
&\leq \sup_{\phi} \lim_{L \to \infty} \int_{\mathcal{X}} \left| \frac{1}{L}\sum_{l=1}^{L} \mathbb{E}_{\mathbf{x}}\phi(\mathbf{X}_{lh}) - \bar{\phi} \right| \tilde{\rho}_h(\mathrm{d}\,\mathbf{x}) \\
&\leq \sup_{\phi} \lim_{L \to \infty} \left( \frac{C_1}{Lh} + C_2 h^K \right) \\
&= Ch^K \ ,
\end{aligned}
\tag{34}
$$

where (34) follows by using the result from Theorem 2. This completes the proof. $\qquad\square$

# G  The Proof of Theorem 5

We separate the proof into proofs for the bias and MSE respectively in the following.

**The proof for the bias:**

*Proof.* Following Theorem 2, in the decreasing step size setting, (20) can be written as:

$$\mathbb{E}\left(\psi(\mathbf{X}_{lh})\right) = \left(\mathbb{I} + h_l \tilde{\mathcal{L}}_l\right)\psi(\mathbf{X}_{(l-1)h}) + \sum_{k=2}^{K} \frac{h_l^k}{k!}\tilde{\mathcal{L}}_l^2 \psi(\mathbf{X}_{(l-1)h}) + O(h_l^{K+1}) \ .$$

Similarly, (21) can be simplified using the step size sequence $(h_l)$ as:

$$
\begin{aligned}
&\mathbb{E}\left(\tilde{\phi} - \bar{\phi}\right) \\
&= \frac{1}{S_L}\left(\mathbb{E}\left(\psi(\mathbf{X}_{Lh})\right) - \psi(\mathbf{X}_0)\right) - \sum_{k=2}^{K}\sum_{l=1}^{L} \frac{h_l^k}{k!S_L}\tilde{\mathcal{L}}_l^k \psi(\mathbf{X}_{(l-1)h}) + O(\frac{\sum_{l=1}^{L} h_l^{K+1}}{S_L})
\end{aligned}
\tag{35}
$$

Similar to the derivation of (24), we can derive the following bounds $k = (2, \cdots, K)$:

$$
\begin{aligned}
\sum_{l=1}^{L} h_l^k \mathbb{E}\tilde{\mathcal{L}}_l^k \psi(\mathbf{X}_{(l-1)h}) &= O\left( \sum_{l=1}^{L}\left( (h_l^{k-1} - h_{l-1}^{k-1})\tilde{\mathcal{L}}_l^{k-1}\psi(\mathbf{X}_{(l-1)h}) + h_l^{K+1} \right) \right) \\
&= O\left( 1 + \sum_{l=1}^{L} h_l^{K+1} \right) \ .
\end{aligned}
\tag{36}
$$

Substitute (36) into (35) and collect low order terms, we have:

$$\mathbb{E}\left(\tilde{\phi} - \bar{\phi}\right) = \frac{1}{S_L}\left(\mathbb{E}\left(\psi(\mathbf{X}_{Lh})\right) - \psi(\mathbf{X}_0)\right) + O(\frac{\sum_{l=1}^{L} h_l^{K+1}}{S_L}) \ . \tag{37}$$

As a result, the bias can be expressed as:

$$\left| \mathbb{E}\tilde{\phi} - \bar{\phi} \right| \leq \left| \frac{1}{S_L} \left( \mathbb{E}\left[ \psi(\mathbf{X}_{Lh}) \right] - \psi(\mathbf{X}_0) \right) + O\left( \frac{\sum_{l=1}^{L} h_l^{K+1}}{S_L} \right) \right|$$

$$\lesssim \left| \frac{1}{S_L} \right| + \left| \frac{\sum_{l=1}^{L} h_l^{K+1}}{S_L} \right) \right|$$

$$= O\left( \frac{1}{S_L} + \frac{\sum_{l=1}^{L} h_l^{K+1}}{S_L} \right) .$$

Taking $L \to \infty$, both terms go to zero by assumption. This completes the proof. $\qquad\square$

**The proof for the MSE:**

*Proof.* Following similar derivations as in Theorem 3, we have that

$$\sum_{l=1}^{L} \mathbb{E}\left( \psi(\mathbf{X}_{lh}) \right) = \sum_{l=1}^{L} \psi(\mathbf{X}_{(l-1)h}) + \sum_{l=1}^{L} h_l \mathcal{L}\psi(\mathbf{X}_{(l-1)h}) + \sum_{l=1}^{L} h_l \Delta V_l \psi(\mathbf{X}_{(l-1)h})$$

$$+ \sum_{k=2}^{K} \sum_{l=1}^{L} \frac{h_l^k}{k!} \tilde{\mathcal{L}}_l^k \psi(\mathbf{X}_{(l-1)h}) + C \sum_{l=1}^{L} h_l^{K+1} .$$

Substitute the Poisson equation (4) into the above equation and divided both sides by $S_L$, we have

$$\hat{\phi} - \bar{\phi} = \frac{\mathbb{E}\psi(\mathbf{X}_{Lh}) - \psi(x_0)}{S_L} + \frac{1}{S_L} \sum_{l=1}^{L-1} \left( \mathbb{E}\psi(\mathbf{X}_{(l-1)h}) + \psi(\mathbf{X}_{(l-1)h}) \right) + \sum_{l=1}^{L} \frac{h_l}{S_L} \Delta V_l \psi(\mathbf{X}_{(l-1)h})$$

$$+ \sum_{k=2}^{K} \sum_{l=1}^{L} \frac{h_l^k}{k! S_L} \tilde{\mathcal{L}}_l^k \psi(\mathbf{X}_{(l-1)h}) + C \frac{\sum_{l=1}^{L} h_l^3}{S_L} .$$

As a result, there exists some positive constant $C$, such that:

$$\mathbb{E}\left( \hat{\phi} - \bar{\phi} \right)^2 \leq C\mathbb{E}\left( \frac{1}{S_L^2} \underbrace{\left( \psi(\mathbf{X}_0) - \mathbb{E}\psi(\mathbf{X}_{Lh}) \right)^2}_{A_1} + \frac{1}{S_L^2} \underbrace{\sum_{l=1}^{L} \left( \mathbb{E}\psi(\mathbf{X}_{(l-1)h}) - \psi(\mathbf{X}_{(l-1)h}) \right)^2}_{A_2} \right.$$

$$\left. + \sum_{l=1}^{L} \frac{h_l^2}{S_L^2} \left\| \Delta V_l \right\|^2 + \underbrace{\sum_{k=2}^{K} \left( \sum_{l=1}^{L} \frac{h_l^k}{k! S_L} \tilde{\mathcal{L}}_l^k \psi(\mathbf{X}_{(l-1)h}) \right)^2}_{A_3} + \left( \frac{\sum_{l=1}^{L} h_l^3}{S_L} \right)^2 \right) \qquad (38)$$

$A_1$ can be bounded by assumptions, and $A_2$ is shown to be bounded by using the fact that $\mathbb{E}\psi(\mathbf{X}_{(l-1)h}) - \psi(\mathbf{X}_{(l-1)h}) = O(\sqrt{h_l})$ from Theorem 3. Furthermore, similar to the proof of Theorem 3, the expectation of $A_3$ can also be bounded by using the formula $\mathbb{E}[\mathbf{X}^2] = (\mathbb{E}\,\mathbf{X})^2 + \mathbb{E}[(\mathbf{X} - \mathbb{E}\,\mathbf{X})^2]$ and (36). It turns out that the resulting terms have order higher than those from the other terms, thus can be ignored in the expression below. After some simplifications, (38) is bounded by:

$$\mathbb{E}\left( \hat{\phi} - \bar{\phi} \right)^2 \lesssim \sum_{l} \frac{h_l^2}{S_L^2} \mathbb{E}\left\| \Delta V_l \right\|^2 + \frac{1}{S_L} + \frac{1}{S_L^2} + \left( \frac{\sum_{l=1}^{L} h_l^{K+1}}{S_L} \right)^2$$

$$= C\left( \sum_{l} \frac{h_l^2}{S_L^2} \mathbb{E}\left\| \Delta V_l \right\|^2 + \frac{1}{S_L} + \frac{(\sum_{l=1}^{L} h_l^{K+1})^2}{S_L^2} \right) \qquad (39)$$

for some $C > 0$, this completes the first part of the theorem. We can see that according to the assumption, the last two terms in (39) approach to 0 when $L \to \infty$. If we further assume $\frac{\sum_{l=1}^{\infty} h_l^2}{S_L^2} = 0$, then the first term in (39) approaches to 0 because:

$$\sum_l \frac{h_l^2}{S_L^2} \mathbb{E} \|\Delta V_l\|^2 \leq \left( \sup_l \mathbb{E} \|\Delta V_l\|^2 \right) \frac{\sum_l h_l^2}{S_L^2} \to 0 .$$

As a result, we have $\lim_{L\to\infty} \mathbb{E} \left( \hat{\phi} - \bar{\phi} \right)^2 = 0$. $\qquad \square$

## H    The Proof of Corollary 6

*Proof.* We use the following inequalities to bound the term $\sum_{l=1}^{L} l^{-\alpha}$:

$$\int_1^L x^{-\alpha} \mathrm{d}x < \sum_{l=1}^{L} l^{-\alpha} < 1 + \int_1^{L-1} x^{-\alpha} \mathrm{d}x .$$

This is easily seen to be true by noting that $\int_l^{l+1} x^{-\alpha} \mathrm{d}x < l^{-\alpha} \times 1 = l^{-\alpha} < \int_{l-1}^{l} x^{-\alpha} \mathrm{d}x$. After simplification, we have

$$\frac{1 - L^{1-\alpha}}{\alpha - 1} < \sum_{l=1}^{L} l^{-\alpha} < \frac{\alpha - (L-1)^{1-\alpha}}{\alpha - 1} . \tag{40}$$

It is then easy to see that the condition for $\sum_{l=1}^{\infty} l^{-\alpha} = \infty$ is $\alpha \leq 1$. Moreover, we notice that other step size assumptions reduce to compare $\sum_{l=1}^{\infty} l^{-\alpha}$ and $\sum_{l=1}^{\infty} l^{-\alpha_1}$ for $\alpha < \alpha_1$, which using (40) has the following bound:

$$\frac{\alpha - 1}{\alpha_1 - 1} \frac{1 - L^{1-\alpha_1}}{\alpha - (L-1)^{1-\alpha}} < \frac{\sum_{l=1}^{L} l^{-\alpha_1}}{\sum_{l=1}^{L} l^{-\alpha}} < \frac{\alpha - 1}{\alpha_1 - 1} \frac{\alpha_1 - (L-1)^{1-\alpha_1}}{1 - L^{1-\alpha}} .$$

As long as $0 < \alpha < 1$ and $\alpha_1 > \alpha$, the above lower and upper bound would approach to 0, thus all the assumptions for the step size sequences are satisfied. $\qquad \square$

## I    On the Euler Integrator and Symmetric Splitting Integrator

### I.1    Euler integrator

We first review the Euler scheme used in SGLD and SGHMC. In SGLD the update for $\mathbf{X}_{lh} (= \boldsymbol{\theta}_{lh})$ follows:

$$\boldsymbol{\theta}_{lh} = \boldsymbol{\theta}_{(l-1)h} - \nabla_{\boldsymbol{\theta}} \tilde{U}_l(\boldsymbol{\theta}_{(l-1)h})h + \sqrt{2h}\boldsymbol{\zeta}_l ,$$

where $h$ is the step size, $\boldsymbol{\zeta}_l$ is a vector of *i.i.d.* standard normal random variables. In SGHMC ($\mathbf{X}_{lh} = (\boldsymbol{\theta}_{lh}, p_{lh})$), it becomes:

$$\boldsymbol{\theta}_{lh} = \boldsymbol{\theta}_{(l-1)h} + \mathbf{p}_{(l-1)h} h, \quad \mathbf{p}_{lh} = (1 - Dh) \mathbf{p}_{(l-1)h} - \nabla_{\boldsymbol{\theta}} \tilde{U}_l(\boldsymbol{\theta}_{(l-1)h})h + \sqrt{2Dh}\boldsymbol{\zeta}_l ,$$

Based on the update equations, it is easily seen that the corresponding Kolmogorov operators $\tilde{P}_h^l$ are

$$\tilde{P}_h^l = e^{h\mathcal{L}_1}, \text{ where } \mathcal{L}_1 \triangleq -\nabla_{\boldsymbol{\theta}} \tilde{U}_l(\boldsymbol{\theta}_{(l-1)h}) \cdot \nabla_{\boldsymbol{\theta}} + 2I : \nabla_{\boldsymbol{\theta}} \nabla_{\boldsymbol{\theta}}^T \tag{41}$$

for SGLD, and

$$\tilde{P}_h^l = e^{h\mathcal{L}_2} \circ e^{h\mathcal{L}_3} , \tag{42}$$

for SGHMC, where $\mathcal{L}_2 \triangleq \mathbf{p} \cdot \nabla_{\boldsymbol{\theta}}$ and $\mathcal{L}_3 \triangleq -D \mathbf{p}_{(l-1)h} \cdot \nabla_{\mathbf{p}} - \nabla_{\boldsymbol{\theta}} \tilde{U}_l(\boldsymbol{\theta}) \cdot \nabla_{\mathbf{p}} + 2DI : \nabla_{\mathbf{p}} \nabla_{\mathbf{p}}^T$.

We show in the following Lemma that the Euler integrator is a 1st-order local integrator.

**Lemma 12.** *The Euler integrator is a 1st-order local integrator, i.e.,*

$$\tilde{P}_h^l = e^{h\tilde{\mathcal{L}}_l} + O(h^2) \, . \tag{43}$$

*Proof.* For the SGLD, according to the Kolmogorov's backward equation (14), for the SGLD, we have

$$\mathbb{E}[f(\boldsymbol{\theta}_{(l-1)h+t})] = e^{t\tilde{\mathcal{L}}_l} f(\boldsymbol{\theta}_{(l-1)h}), \qquad 0 \le t \le h \, , \tag{44}$$

where $\tilde{\mathcal{L}}_1 \triangleq -\nabla_{\boldsymbol{\theta}}\tilde{U}_l(\boldsymbol{\theta}) \cdot \nabla_{\boldsymbol{\theta}} + 2I : \nabla_{\boldsymbol{\theta}}\nabla_{\boldsymbol{\theta}}^T$. Note $\tilde{U}_l(\boldsymbol{\theta})$ can be expanded by Taylor's expansion to the 1st-order such that (based on $\boldsymbol{\theta}_{lh} = \boldsymbol{\theta}_{(l-1)h} + O(h)$):

$$\tilde{\mathcal{L}}_1 = -\nabla_{\boldsymbol{\theta}}\tilde{U}_l(\boldsymbol{\theta}_{(l-1)h}) \cdot \nabla_{\boldsymbol{\theta}} + 2I : \nabla_{\boldsymbol{\theta}}\nabla_{\boldsymbol{\theta}}^T + O(h)$$
$$= \mathcal{L}_1 + O(h) \, .$$

Substituting the above into (44) and use the definition (41), we have

$$\tilde{P}_h^l = e^{h\tilde{\mathcal{L}}_l} + O(h^2) \, .$$

For the SGHMC, following similar derivations, we have:

$$\mathcal{L}_2 = \tilde{\mathcal{L}}_2 \to e^{h\mathcal{L}_2} = e^{h\tilde{\mathcal{L}}_2} + O(h^2) \, ,$$
$$e^{h\mathcal{L}_3} = e^{h\tilde{\mathcal{L}}_3} + O(h^2) \, ,$$

where $\tilde{\mathcal{L}}_2 \triangleq \mathbf{p} \cdot \nabla_{\boldsymbol{\theta}}$ and $\tilde{\mathcal{L}}_3 \triangleq -D\mathbf{p}\, h \cdot \nabla_{\mathbf{p}} - \nabla_{\boldsymbol{\theta}}\tilde{U}_l(\boldsymbol{\theta}) \cdot \nabla_{\mathbf{p}} + 2DI : \nabla_{\mathbf{p}}\nabla_{\mathbf{p}}^T$ are the splitting for the true generator $\tilde{\mathcal{L}}_l$.

Now using the Baker–Campbell–Hausdorff (BCH) formula, we have

$$e^{h\mathcal{L}_2} \circ e^{h\mathcal{L}_3} = e^{h\tilde{\mathcal{L}}_2} \circ \left( e^{h\tilde{\mathcal{L}}_3} + O(h^2) \right)$$
$$= e^{h(\tilde{\mathcal{L}}_2 + \tilde{\mathcal{L}}_3) + O(h^2)} + O(h^2) = e^{h\tilde{\mathcal{L}}_l} + O(h^2)$$

As a result, $\tilde{P}_h^l = e^{h\tilde{\mathcal{L}}_l} + O(h^2)$ for SGHMC. $\qquad\square$

## I.2 Symmetric splitting integrator

In symmetric splitting scheme, the generator $\tilde{\mathcal{L}}_l$ is split into a couple of sub-generators which can be solved analytically. For example, in SGHMC, it is split into: $\tilde{\mathcal{L}}_l = \mathcal{L}_A + \mathcal{L}_B + \mathcal{L}_{O_l}$, where

$$\mathcal{A} \triangleq \mathcal{L}_A = \mathbf{p} \cdot \nabla_{\boldsymbol{\theta}}, \ \ \mathcal{B} \triangleq \mathcal{L}_B = -D\mathbf{p} \cdot \nabla_{\mathbf{p}}, \ \ \mathcal{O}_l \triangleq \mathcal{L}_{O_l} = -\nabla_{\boldsymbol{\theta}}\tilde{U}_l(\boldsymbol{\theta}) \cdot \nabla_{\mathbf{p}} + 2D : \nabla_{\mathbf{p}}\nabla_{\mathbf{p}}^T \, .$$

These sub-generators correspond to the following analytically solvable SDEs:

$$A : \begin{cases} \mathrm{d}\boldsymbol{\theta} &= \mathbf{p}\,\mathrm{d}t \\ \mathrm{d}\mathbf{p} &= 0 \end{cases} , B : \begin{cases} \mathrm{d}\boldsymbol{\theta} &= 0 \\ \mathrm{d}\mathbf{p} &= -D\mathbf{p}\,\mathrm{d}t \end{cases} , O : \begin{cases} \mathrm{d}\boldsymbol{\theta} &= 0 \\ \mathrm{d}\mathbf{p} &= -\nabla_{\boldsymbol{\theta}}\tilde{U}_l(\boldsymbol{\theta})\mathrm{d}t + \sqrt{2D}\mathrm{d}W \end{cases}$$

Based on the splitting, the Kolmogorov operator $\tilde{P}_h^l$ can be seen to be:

$$\tilde{P}_h^l \triangleq e^{\frac{h}{2}\mathcal{L}_A} \circ e^{\frac{h}{2}\mathcal{L}_B} \circ e^{h\mathcal{L}_{O_l}} \circ e^{\frac{h}{2}\mathcal{L}_B} \circ e^{\frac{h}{2}\mathcal{L}_A},$$

We show that the corresponding integrator is a 2nd-order local integrator below.

**Lemma 13.** *The symmetric splitting integrator is a 2nd-order local integrator, i.e.,*

$$\tilde{P}_h^l = e^{h\tilde{\mathcal{L}}_l} + O(h^3) \, . \tag{45}$$

*Proof.* This follows from direct calculation using the BCH formula. Specifically,

$$e^{\frac{h}{2}\mathcal{A}}e^{\frac{h}{2}\mathcal{B}} = e^{\frac{h}{2}\mathcal{A}+\frac{h}{2}\mathcal{B}+\frac{h^2}{8}[\mathcal{A},\mathcal{B}]+\frac{1}{96}([\mathcal{A},[\mathcal{A},\mathcal{B}]]+[\mathcal{B},[\mathcal{B},\mathcal{A}]])+\cdots} \tag{46}$$

$$= e^{\frac{h}{2}\mathcal{A}+\frac{h}{2}\mathcal{B}+\frac{h^2}{8}[\mathcal{A},\mathcal{B}]} + O(h^3) \, , \tag{47}$$

where $[X,Y] \triangleq XY - YX$ is the commutator of $X$ and $Y$, (46) follows from the BCH formula, and (47) follows from Assumption 1 such that the remainder high order terms are bounded [24], so the error term $O(h^3)$ can be taken out from the exponential map using Taylor expansion. Similarly, for the other composition, we have

$$e^{h\mathcal{O}_l} e^{\frac{h}{2}\mathcal{A}} e^{\frac{h}{2}\mathcal{B}} = e^{h\mathcal{O}_l} \left( e^{\frac{h}{2}\mathcal{A} + \frac{h}{2}\mathcal{B} + \frac{h^2}{8}[\mathcal{A},\mathcal{B}]} + O(h^3) \right)$$

$$= e^{h\mathcal{O}_l + \frac{h}{2}\mathcal{A} + \frac{h}{2}\mathcal{B} + \frac{h^2}{8}[\mathcal{A},\mathcal{B}] + \frac{1}{2}[h\mathcal{O}_l, \frac{h}{2}\mathcal{A} + \frac{h}{2}\mathcal{B} + \frac{h^2}{8}[\mathcal{A},\mathcal{B}]]} + O(h^3)$$

$$= e^{h\mathcal{O}_l + \frac{h}{2}\mathcal{A} + \frac{h}{2}\mathcal{B} + \frac{h^2}{8}[\mathcal{A},\mathcal{B}] + \frac{h^2}{4}[\mathcal{O}_l,\mathcal{A}] + \frac{h^2}{4}[\mathcal{O}_l,\mathcal{B}]} + O(h^3)$$

$$e^{\frac{h}{2}\mathcal{A}} e^{h\mathcal{O}_l} e^{\frac{h}{2}\mathcal{A}} e^{\frac{h}{2}\mathcal{B}} = e^{\frac{h}{2}\mathcal{A}} \left( e^{h\mathcal{O}_l + \frac{h}{2}\mathcal{A} + \frac{h}{2}\mathcal{B} + \frac{h^2}{8}[\mathcal{A},\mathcal{B}] + \frac{h^2}{4}[\mathcal{O}_l,\mathcal{A}] + \frac{h^2}{4}[\mathcal{O}_l,\mathcal{B}]} + O(h^3) \right)$$

$$= e^{h\mathcal{O}_l + h\mathcal{A} + \frac{h}{2}\mathcal{B} + \frac{h^2}{4}[\mathcal{A},\mathcal{B}] + \frac{h^2}{2}[\mathcal{O}_l,\mathcal{B}]} + O(h^3)$$

$$\tilde{P}_h^l \triangleq e^{\frac{h}{2}\mathcal{B}} e^{\frac{h}{2}\mathcal{A}} e^{h\mathcal{Z}} e^{\frac{h}{2}\mathcal{A}} e^{\frac{h}{2}\mathcal{B}} = e^{\frac{h}{2}\mathcal{B}} \left( e^{h\mathcal{O}_l + h\mathcal{A} + \frac{h}{2}\mathcal{B} + \frac{h^2}{4}[\mathcal{A},\mathcal{B}] + \frac{h^2}{2}[\mathcal{O}_l,\mathcal{B}]} + O(h^3) \right)$$

$$= e^{h\mathcal{O}_l + h\mathcal{A} + h\mathcal{B} + \frac{h^2}{4}[\mathcal{A},\mathcal{B}] + \frac{h^2}{2}[\mathcal{O}_l,\mathcal{B}] + \frac{h^2}{4}[\mathcal{B},\mathcal{A}] + \frac{h^2}{4}[\mathcal{B},\mathcal{O}_l] + \frac{h^2}{8}[\mathcal{B},\mathcal{B}]} + O(h^3)$$

$$= e^{h(\mathcal{B} + \mathcal{A} + \mathcal{O}_l)} + O(h^3)$$

$$= e^{h(\mathcal{L} + \Delta V_l)} + O(h^3) = e^{h\tilde{\mathcal{L}}_l} + O(h^3) .$$

This completes the proof. $\qquad \square$

## J  Mean Flow Error Analysis

In addition to the finite time ergodic error studied previously, we study the mean flow error in this section. To this end, we first define the exact mean flow to be the solution operator of the Kolmogorov's backward equation $\mathbb{E}[f(\mathbf{X}_T)] = e^{T\mathcal{L}} f(\mathbf{X}_0)$ over time $T = Lh$, i.e., $\varphi_T \triangleq e^{T\mathcal{L}}$. With our splitting method with stochastic gradients for each minibatch, the mean flow operator consists of a composition of $L$ local mean flows, i.e., $\tilde{\varphi}_T^h \triangleq e^{h\mathcal{L}_L} \circ \cdots \circ e^{h\mathcal{L}_1} \triangleq \circ_{l=1}^{L} e^{h\mathcal{L}_l}$, each coming from a minibatch. Our goal in this section is to compare $\varphi_T$ with $\tilde{\varphi}_T^h$. When the underlying equations of motion are PDEs, i.e., no Brownian motion like the Hamiltonian PDE, $\varphi_T(\mathbf{X}_0)$ corresponds to the exact solution trajectory of the PDE, whereas $\tilde{\varphi}_T^h$ is the trajectory of splitting methods with stochastic gradients. [7] shows that in this case $\varphi_T(\mathbf{X}_0)$ is not close to $\tilde{\varphi}_T^h$ in general. In the section we extend this result by showing that the conclusion also holds in the SDE case. We comment that this result is not as surprising as pointed out in [7] because as pointed out in the introduction, such sample wise convergence is not interesting in most real applications.

**Theorem 14.** *In SGHMC with the symmetric splitting integrator, the difference between the stochastic mean flow operator $\tilde{\varphi}_T^h$ and the exact flow operator $\varphi_T$ depends on the running time $T$ and stochastic gradients in each minibatch, and is given by the following formula,*

$$\left\| \tilde{\varphi}_T^h - \varphi_T \right\| = C \left\| \frac{1}{L} \sum_{l=1}^{L} \Delta V_l + h \left( [\mathcal{L}, \Delta V_1] + [\mathcal{L}, \Delta V_L] \right) \right\| T + O(h^2) ,$$

*for some positive constant $C$.*

We can see from Theorem 14 that $\tilde{\varphi}_T^h$ is not close to $\varphi_T$ because of the uncontrollable terms $\Delta V_l$ with stochastic gradients, thus SG-MCMCs are not sample-wise convergence.

*Proof.* First, applying Kolmogorov's backward equation on the original SDE (1) with generator $\mathcal{L}$, the true mean flow $\varphi_T(\mathbf{X}_0)$ can be expressed as:

$$\varphi_T(\mathbf{X}_0) = e^{T\mathcal{L}}(\mathbf{X}_0) . \tag{48}$$

Now we want to compute the mean flow of the splitting scheme: $\circ_{l=1}^{L} \hat{\varphi}_{lh}(\mathbf{X}_0)$. We will split the SDE into several parts, with the Brownian motion term going with the stochastic gradient term. To shown the proof on a different SG-MCMC algorithm, we use the SGHMC with Riemannian information

geometry (SGRHMC) defined below. Other stochastic gradient MCMC follows similarly. For the SGRHMC, we have

$$
d \begin{bmatrix} \boldsymbol{\theta} \\ \mathbf{p} \end{bmatrix} = \underbrace{\begin{bmatrix} 0 \\ -\left(\nabla_{\boldsymbol{\theta}} U(\boldsymbol{\theta}) + \frac{1}{2}\nabla_{\boldsymbol{\theta}} \log \det G(\boldsymbol{\theta})\right) dt + \sqrt{2D} dW \end{bmatrix}}_{B}
$$

$$
+ \underbrace{\begin{bmatrix} 0 \\ -DG(\boldsymbol{\theta})^{-1} \mathbf{p} \end{bmatrix} dt}_{A} + \underbrace{\begin{bmatrix} G(\boldsymbol{\theta})^{-1} \mathbf{p} \\ \nu(\boldsymbol{\theta}, \mathbf{p}) \end{bmatrix} dt}_{O} \tag{49}
$$

The splitting scheme we consider is the BAOAB scheme. Denote

$$
\mathcal{B} = \mathcal{L}_B = -\left(\nabla_{\boldsymbol{\theta}} U(\boldsymbol{\theta}) + \frac{1}{2}\nabla_{\boldsymbol{\theta}} \log \det G(\boldsymbol{\theta})\right) \cdot \nabla_{\mathbf{p}} + 2D\triangle_{\mathbf{p}}
$$

$$
\mathcal{A} = \mathcal{L}_A = -DG^{-1}\mathbf{p} \cdot \nabla_{\mathbf{p}}
$$

$$
\mathcal{O} = \mathcal{L}_O = G^{-1}\mathbf{p} \cdot \nabla_{\boldsymbol{\theta}} + \nu \cdot \nabla_{\mathbf{p}} .
$$

Note that $\mathcal{L} = \mathcal{A} + \mathcal{B} + \mathcal{O}$. In the stochastic gradient case, we are using the stochastic gradient from the $l$-th minibatch in the splitting scheme, thus we need to modify the operator $\mathcal{B}$ as:

$$
\mathcal{B}_l \triangleq \mathcal{L}_{\mathcal{B}_l} = -\left(\nabla_{\boldsymbol{\theta}} \tilde{U}_l(\boldsymbol{\theta}) + \frac{1}{2}\nabla_{\boldsymbol{\theta}} \log \det G(\boldsymbol{\theta})\right) \cdot \nabla_{\mathbf{p}} + 2D \triangle_{\mathbf{p}} ,
$$

where $\nabla_{\boldsymbol{\theta}} \tilde{U}_l$ is evaluated on a subset of data. We emphasis the notation that $\Delta V_l \triangleq \mathcal{B}_l - \mathcal{B} = \left(\nabla_{\boldsymbol{\theta}} \tilde{U}_l - \nabla_{\boldsymbol{\theta}} U\right) \cdot \nabla_{\mathbf{p}}$, it can be shown that $\Delta V_l$ commutes with each other, e.g., $\Delta V_i \Delta V_j = \Delta V_j \Delta V_i$.

We know from Section I.2 that using the symmetric splitting integrator, the mean flow $\tilde{\varphi}_h^l$ is close to $e^{h(\mathcal{L}+\Delta V_l)}$ with a $O(h^3)$ error, i.e.,

$$
\varphi_h^l = e^{h(\mathcal{L}+\Delta V_l)} + O(h^3) .
$$

Similar to the proof of the symmetric splitting error, we can calculate the composition of the mean flows for two mini-batches $i$ and $j$ using the BCH formula as:

$$
\tilde{\varphi}_h^j \circ \tilde{\varphi}_h^i = e^{h(\mathcal{B}+\mathcal{A}+\mathcal{O})+h\Delta V_i} \circ e^{h(\mathcal{B}+\mathcal{A}+\mathcal{Z})+h\Delta V_j} + O(h^3)
$$

$$
= e^{2h\mathcal{L}+h(\Delta V_i+\Delta V_j)+\frac{h^2}{2}[\mathcal{L}+\Delta V_j, \mathcal{L}+\Delta V_i]} + O(h^3)
$$

$$
= e^{2h\mathcal{L}+h(\Delta V_i+\Delta V_j)+\frac{h^2}{2}([\mathcal{L},\Delta V_i]+[\Delta V_j,\mathcal{L}])} + O(h^3) ,
$$

where we have used the fact that $\{\Delta V_i\}$ commutes with each other to cancel out the $[\Delta V_i, \Delta V_j]$ term in the BCH formula. Similarly, for the first three mini-batches $i, j, k$, we have

$$
\tilde{\varphi}_h^k \circ \tilde{\varphi}_h^j \circ \tilde{\varphi}_h^i = e^{h(\mathcal{B}+\mathcal{A}+\mathcal{O})+h\Delta V_i} \circ e^{h(\mathcal{B}+\mathcal{A}+\mathcal{O})+h\Delta V_j} + O(h^3)
$$

$$
= e^{3h\mathcal{L}+h(\Delta V_i+\Delta V_j+\Delta V_k)+h^2([\mathcal{L},\Delta V_i]+[\mathcal{L},\Delta V_k])} + O(h^3) .
$$

Similarly, we can do the composition for the entire trajectory, resulting in after simplification:

$$
\circ_{l=1}^L \tilde{\varphi}_h^l = e^{(Lh)\mathcal{L}+(Lh)\frac{1}{L}\sum_{l=1}^L \Delta V_l+(Lh)h([\mathcal{L},\Delta V_1]+[\mathcal{L},\Delta V_L])} + (Lh)O(h^2)
$$

$$
= e^{T\mathcal{L}+T\frac{1}{L}\sum_{l=1}^L \Delta V_l+Th([\mathcal{L},\Delta V_1]+[\mathcal{L},\Delta V_L])} + O(h^2) \tag{50}
$$

This completes the first part of the theorem. From Assumption 1, we can expand and bound (50) with the step size $h$ for finite time $T$ as:

$$
\tilde{\varphi}_T(\mathbf{X}_0) = \left(T\mathcal{L} + T\frac{1}{L}\sum_{l=1}^L \Delta V_l + Th\left([\mathcal{L}, \Delta V_1] + [\mathcal{L}, \Delta V_L]\right)\right)(\mathbf{X}_0) + O(h^2) .
$$

Figure 5: *Bias* and *MSE* for SGHMC with different step size rates.

Similarly, for the true mean flow $\varphi_T(\mathbf{X}_0)$, it is easy to get

$$\varphi_T(\mathbf{X}_0) = \underbrace{e^{h\mathcal{L}} \circ e^{h\mathcal{L}} \circ \cdots \circ e^{h\mathcal{L}}}_{L} = T\mathcal{L}(\mathbf{X}_0) + O(h^2) \,.$$

As a result:

$$\|\varphi_T(\mathbf{X}_0) - \tilde{\varphi}_T(\mathbf{X}_0)\| = \left\|\left(T\frac{1}{L}\sum_{l=1}^{L}\Delta V_l + Th\left([\mathcal{L}, \Delta V_1] + [\mathcal{L}, \Delta V_L]\right)\right)(\mathbf{X}_0) + O(h^2)\right\|$$

$$= \left\|\left(\sum_{l=1}^{L}\Delta V_l + T\left([\mathcal{L}, \Delta V_1] + [\mathcal{L}, \Delta V_L]\right)\right)(\mathbf{X}_0)\right\|h + O(h^2)$$

$$= C\left\|\frac{1}{L}\sum_{l=1}^{L}\Delta V_l + h\left([\mathcal{L}, \Delta V_1] + [\mathcal{L}, \Delta V_L]\right)\right\|T + O(h^2)$$

This completes the proof. □

## K  Additional Experiments

### K.1  Synthetic data

We plot the traces of bias and MSE with step size $h \propto L^\alpha$ for different rates $\alpha$ in Figure 5. We can see that when the rates are smaller than the theoretically optimal bias rates $\alpha = -1/3$ and MSE rate $\alpha = -1/5$, the bias and MSE tend to decrease faster than the optimal rates at the beginning, but eventually they slow down and are surpassed by the optimal rates. This on the other hand suggests if only a small number of iterations were available in the SG-MCMCs, setting a larger step size than the theoretically optimal one might be beneficial in practice.

In addition, Figure 6 shows a comparison of the bias and MSE for SGHMC and SGLD. The step sizes are set to $h = CL^{-\alpha}$, with $\alpha$ choosing according to the theory for SGLD and SGHMC respectively. To be fair, the constants $C$ are selected via a grid search from 1e-3 to 0.5 with an interval of 2e-3 for $L = 200$, it is then fixed during other $L$ values. The parameter $D$ in SGHMC is selected within $(10, 20, 30)$ as well. As indicated by both our theorems and experiments, SGHMC endows a much faster convergence speed than SGHMC on both the bias and MSE.

Figure 7 plots the traces of bias and MSE with decreasing step sizes $h \propto l^\alpha$ for different rates $\alpha$ in the same Gaussian model. Again we can see that the optimal decreasing rates agree with the theory. Figure 8 shows a comparison of bias and MSE for SGHMC and SGLD with decreasing step sizes $h \propto l^{-\alpha}$ on the same Gaussian model. We follow the same procedure as in Section 3.1 to select

(a) Bias

(b) MSE

Figure 6: Comparisons of *bias* and *MSE* for SGHMC and SGLD on a simple Gaussian model.

(a) Bias

(b) MSE

Figure 7: *Bias* and *MSE* for decreasing step size SGHMC with different step size rates.

parameters for SGLD and SGHMC. Specifically, the decreasing rate parameter $\alpha$ is set to $1/2$ and $1/3$ in SGLD and SGHMC for the bias, $1/3$ and $1/5$ for the MSE. We can see that SGHMC still obtain a faster convergence speed, though the benefit is not as large as using fix step size.

## K.2 LDA & SBN

We first the list quantitative results of the LDA and SBN models in Table 1. It is clear that in both models the SGHMC is much better than the SGLD due to the introduction of momentum variables in the dynamics (similar to the SGD with momemtum [1] in the optimization literature); and the splitting integrator also works better than the Euler integrator due to the higher order errors in splitting integrators. For a fair comparison, we did not consider a better version of the SGLD with Riemannian information geometry of posterior distributions on probabilistic simplexes [19].

Table 1: Comparisons for different algorithms. $K$ in LDA means #topics, $J$ in SBN means #hidden units; suffix 'S' means the *symmetric splitting integrator*, 'E' means the *Euler integrator*.

| LDA (Test perplexity) | | | | SBN (Test neg-log-likelihood) | | | |
|---|---|---|---|---|---|---|---|
| K | SGHMC-S | SGHMC-E | SGLD-E | J | SGHMC-S | SGHMC-E | SGLD-E |
| 200 | **1168** | 1180 | 2496 | 100 | **103** | 105 | 126 |
| 500 | **1157** | 1187 | 2511 | 200 | **98** | 100 | 110 |

(a) Bias

(b) MSE

Figure 8: Comparisons of *Bias* and *MSE* for SGHMC and SGLD with decreasing step sizes on a simple Gaussian model.

Next a plot of the test perplexities decreasing with the number of documents processed for the whole dataset is given in Figure 9 (top), for a comparison of the Euler integrator and the proposed symmetric splitting integrator. We can see that the symmetric splitting integrator decreases faster than the Euler integrator. Furthermore, the dictionary learned by the SGHMC with the symmetric splitting integrator is also given in Figure 9 (bottom).

Figure 9: Top: comparisons of Splitting and Euler methods in LDA. Bottom: Dictionary learned by SGHMC in SBN.