[Reviews · NeurIPS 2015]

Submitted by Assigned_Reviewer_1

Quality Formal analysis of algorithms is important and this is a welcome addition to the NIPS literature. The tools used are quite sophisticated for a NIPS audience but this is a good thing in strengthening the contributions in this area.

Clarity This will be a tough paper for most people at NIPS to read and understand. nevertheless it is the sort of work that should be encouraged and with the appropriate background this work is well written and clearly presented.

Originality The analysis provides formal justification for what could be considered as ad-hoc algorithm constructions - this is a good thing - and as such is novel.

Significance Difficult to say what the impact will be - nevertheless I would advocate this being included in the conference.
Summary: An important piece of formal analysis. The conclusions seem natural - rates of convergence of SG schemes are improved with higher order integrators. This seems natural but not obvious so the analysis is welcome addition to this literature. I am familiar with the analysis provided in a different context related to discrete probabilistic solutions of ODEs so appreciate this being presented at NIPS.

Submitted by Assigned_Reviewer_2

This a logical and not unexpected development in the "en vogue" area of SGLD based MCMC algorithms, and the manuscript inspires confidence. The main idea is to use recent developments and theoretical results from the molecular simulation literature and apply them to some types of Bayesian problems. As expected the approach performs betters (since the integrators are higher order). One thing on which I could not see any comment in the manuscript is computational cost i.e. do the higher order algorithms still outperform others in terms of e.g. mean square error, at fixed computational cost?
Summary: Someone had to do this and the manuscript inspires confidence that the authors know what they are doing.

Submitted by Assigned_Reviewer_3

The paper presents a convergence analysis for stochastic gradient Langevin algorithms. The results are very encouraging, since they show that it is possible to improve the rate of convergence by using high-order integrators. Indeed, previous convergence analysis for the SGLD algorithm have been rather discouraging, since the best possible rate in MSE has been shown to be L^{-2/3} (compared to the standard L^{-1} Monte Carlo rate). Guided by the new results, the authors also propose a 2nd order slitting integrator which attains a rate of L^{-4/5}, i.e. much closer to the standard Monte Carlo rate than SGLD (but still not quite there).

I admit to not having checked the proofs in the supplementary material in detail, but the theory looks solid. I have a few questions and suggestions for improvement:

1. You state in the introduction that you carry out the analysis for general SG-MCMC algorithms, but I do not understand what you mean by this. Is not the analysis in fact rather tailored to numerically integrated stochastic versions of the second order Langevin system? In particular, the way you define $\tilde L_l$ and $\Delta V_l$ seems to be specific to the 2nd order Langevin system. For instance, in the SG-HMC algorithm by Chen et al. 2014 [1] they use different scaling matrices for the drift and the noise terms, and it thus seems to me that this algorithm does not quite fit into the framework that you consider. Does your analysis apply to this algorithm? Likewise, the SG-NHT algorithm by Ding et al. 2014 [2] is based on a different, extended, system and it is not completely obvious that this fits into your framework. I think that you should be more specific about the precise requirements that you have on the diffusion (1) for your results to hold.

2. In the proposed integrator, systems B and O_l together define an Ornstein-Uhlenbeck process for p (\theta is kept fixed). Why do you propose to split this into B and O_l, rather than integrating exactly the OU process? (This would also lead to a symmetric integrator, ACA, where C=B+O_l.)

3. I found Section 5 "large-scale machine learning applications" too brief to provide any valuable insights. I would suggest that you focus on one of the two applications in the main paper, and move the second application completely to the supplementary material.

4. Can you comment on the possibility of using other (i.e., not splitting-based) high-order integrators for SG-HMC and SGLD? Are there any standard integrators that could be used to obtain a L^{-4/5} rate for SGLD?

*** Minor points/typos:

5. Line 121. Are you missing a minus sign in the definition of V_l (since \grad_\theta U comes with a negative sign in the definition of F)?

6. Line 261. The condition should be $\sum_{l=1}^L h_l^2 / S_L^2 \rightarrow 0$, no?

7. What is L for the fixed step-size results presented in Figure 2? Is it still 10^6 (as in Figure 1) or do you use a smaller value here?

8. Line 397: Should be $L = 100$K and not $L = 100K$ (or even better, $L = 10^5$).
Summary: The paper presents a convergence theory for stochastic gradient Langevin algorithms. The key point with the new result is that it is possible to improve the rate of such algorithms by using high-order integrators, and a 2nd order splitting method is proposed which improves the rate compared to the popular SGLD method.

Submitted by Assigned_Reviewer_4

This paper analyses the role of numerical integrators in SGMCMC methods.

The analysis shows how the order (accuracy) of a numerical integration method influences the convergence results for SGMCMC, along the way providing a general weak convergence result which does not depend on the specific choice of numerical integrator.

The analysis also shows that the error in the stationary invariant measure of SGMCMC is influenced by the choice of integrator more than by the errors induced by the stochastic gradient approximation.

Finally the paper shows that with a suitable choice of decreasing step-size schedule, SGMCMC is consistent (ie, asymptotically unbiased and with decreasing error).

The theoretical results are used as motivation to develop a new numerical integrator of higher order, based on a splitting scheme.

The result is a 2nd order symmetric integrator compared to the 1st order Euler integrators which are typically used.

The theory is verified on synthetic data and the proposed integrator is compared against 1st order Euler on synthetic and real datasets.

The results show that the proposed integrator has consistently smaller bias and is more robust to larger step sizes.

Overall this is a very strong paper and I have little to add to it.

One thing which I think does need addressing is the connection to the traditional leapfrog integrator used in HMC and elsewhere.

The proposed integrator has a very similar feel to this method and this should be discussed, even if briefly, in the paper.
Summary: The paper provides theoretical convergence results for an important class of MCMC methods.

These results provide for new methods which are demonstrated in synthetic and real experiments.

Overall the paper seems well written and the results are significant.

Submitted by Assigned_Reviewer_5

The authors develop theory to analyze finite-time ergodic error of general SG-MCMC as well as the asymptotic invariant measure.

With their theory, 2-order integrator SG-MCMC are also proposed for robustness. Experiments validate their theory and their proposed methods.

The paper is with strong theoretical support and the results are original and significant.
Summary: Theories for SG-MCMC convergence are proposed. The paper is original and significant.

Author Feedback
Author rebuttal: We thank all reviewers for their valuable comments. We address specific points below and will add corresponding clarifications in our next revision.

Reviewer 2:
A1: Our analysis can be applied in all types of Ito diffusion with two sources of approximating errors: from the numerical integrator and from the stochastic gradients. SGLD, SGHMC, and SGNHT all belong to such Ito diffusion. We will provide their Ito diffusion formulations in our supplementary material.

$\tilde{L}_I$ is defined as the approximate generator (using SG instead of gradient) for any Ito diffusions, where its approximation error is defined as $\Delta V_l$. Although in line 121 and 174, we used $\Delta V_l$ from the 2nd-order LD (which is not needed for line 174), both definitions and the analysis are not limited to 2nd-order LD. We will make it clearer in our next revision.

A2: For the 2nd-order Langevin systems, it is indeed possible to combine B and O_I into one O-U process and integrate it exactly. Thank you for bringing this to our attention! For some other SDEs (e.g. the one for SGNHT), it may be hard to find analytical solutions for combined B and O_I. Our Lemma 8 shows that the splitting scheme will still be 2nd-order without combining B and O_I, thus is expected to have similar performances.

A3: This is a valuable suggestion and we will consider it in our next revision. Thank you!

A4: In general, simple heuristic generalizations of high-order integrators in ODEs (e.g. Runge-Kutta methods) do not necessarily lead to high-order methods in SDEs. Other high-order integrators in SDEs such as high-order Taylor methods (see eg. [r1]) may require evaluations of high-order derivatives of F(X). Regardless of their computational expense, they also fit into our theory with a modified $\Delta V_l$.

A5-8: Thanks for the minor points, we will revise them accordingly.

Reviewer 3:
Q: How is the symmetric splitting scheme connected to the leapfrog scheme in HMC?
A: The symmetric splitting scheme (in SDEs) can be considered to be a generalization of the leapfrog scheme (which is also 2nd-order) in standard ODEs. We will discuss this in the next revision. Thank you!

Reviewer 6:
Q: What is the computational cost of the 2nd-order integrator compared to that of the 1st-order method?
A: The computational cost of the 2nd order integrator is about the same as the 1st order Euler scheme (see the update equations in lines 302--304), because the majority of the cost lies in evaluating the stochastic gradient.

[r1] P. E. Kloeden, E. Platen, "Numerical Solution of Stochastic Differential Equations", Springer, 1992.